## RESEARCH ARTICLE

# RhoGEF2 overexpression induces cell competition dependent on Ptp10D, Crumbs and the Hippo signaling pathway

Natasha Fahey-Lozano[1], Marta Portela[1], John E. La Marca[1,2,3,4] and Helena E. Richardson[1,*]

## ABSTRACT

In *Drosophila melanogaster* larval epithelial tissues, cells containing mutations in the apico-basal polarity protein, Scrib, are eliminated by cell competition when surrounded by wild-type cells. In *scrib* mutant cells, signaling mediated by the receptor-type tyrosine phosphatase Ptp10D upon engagement with its ligand Sas in the surrounding wild-type cells triggers cell competition via EGFR-Ras pathway inhibition and JNK pathway activation, which induces apoptosis of the mutant cells. Here, we investigate whether overexpression of *RhoGEF2* (*RhoGEF2^OE^*), which induces Rho signaling and affects actin cytoskeleton regulators, acts similarly to Scrib depletion in cell competition. We show that *RhoGEF2^OE^* cells are eliminated when surrounded by wild-type cells and that *Ptp10D* knockdown increases *RhoGEF2^OE^* clone growth. Mechanistically, in clones moderately overexpressing *RhoGEF2^OE^*, *Ptp10D* knockdown rescued cell elimination by reducing Hippo signaling. Additionally, mutations in the apical cell polarity protein, Crb, partially rescued the elimination of *RhoGEF2^OE^* clones. In this setting, in which *RhoGEF2^OE^* is highly overexpressed, JNK and Hippo signaling were elevated whereas EGFR-Ras signaling was reduced, and *crb* loss normalized these pathways. Thus, *RhoGEF2^OE^* leads to clone elimination dependent on Crb, Ptp10D and Hippo signaling.

KEY WORDS: ***Drosophila*, Cell competition, RhoGEF2, Ptp10D, Crumbs, Hippo**

## INTRODUCTION

Cell competition is an intrinsic surveillance mechanism that functions within tissues, where cells gauge the fitness of their neighbors. The 'more fit' cells ('winners') instigate the elimination of 'less fit' ('loser') cells (Morata and Ripoll, 1975; Simpson and Morata, 1981). Importantly, the elimination of these defective (or suboptimal) loser cells is context dependent, such that the loser cells are completely viable when surrounded by cells of the same loser genotype (Morata and Ripoll, 1975; Wodarz, 2000). After cell competition occurs, any space previously occupied by now eliminated loser cells is then filled by winner cells through the process of 'compensatory cell proliferation', maintaining tissue homeostasis (Fan and Bergmann, 2008; Martín et al., 2009). It is now clear that cell competition mechanisms play an important role during development, aging and cancer (Merino et al., 2016).

Cell competition is involved in the elimination of cells with mutations in the core regulators of apico-basal cell polarity, such as *scribble* (*scrib*), *discs large 1* (*dlg1*, hereafter *dlg*) and *lethal (2) giant larvae* [*l(2)gl*, hereafter *lgl*], which occurs via JNK-dependent apoptosis (Brumby and Richardson, 2003; Chen et al., 2010; Igaki et al., 2009; Menendez et al., 2010; Ohsawa et al., 2011; Tamori et al., 2010). Similarly, clones overexpressing the apical polarity determinant *crumbs* (*crb*) are losers, and are eliminated by apoptosis when surrounded by wild-type cells (Hafezi et al., 2012). In contrast, *crb* mutant clones are winners, undergoing increased proliferation and inducing apoptosis in the surrounding wild-type cells (Hafezi et al., 2012). *scrib*, *dlg* and *lgl* are also recognized as tumor suppressor genes, as mutations in these genes can cause entire tissues to become tumorigenic (Wodarz, 2000) and therefore cell competition involving these genes (and other tumor suppressors) is also known as 'tumor suppressive cell competition' (Kanda and Igaki, 2020).

Hippo signaling is a central pathway for cell competition mechanisms, including those involving *crb* and tumor-suppressive cell competition (Chen et al., 2012; Froldi et al., 2010; Hafezi et al., 2012; Menendez et al., 2010; Tyler et al., 2007). This pathway is important in regulating tissue growth and has been implicated in various cancers (Staley and Irvine, 2012; Zhao et al., 2011). The core components of the Hippo pathway include the kinases Hippo and Warts, along with the adaptors Salvador and Mats, which together prevent the nuclear accumulation of the co-transcription factor Yorkie (Yki; known as YAP1/2 in mammals). When the Hippo pathway is downregulated, Yki accumulates in the nucleus and binds to the TEAD family transcription factor Scalloped (Sd), leading to the transcription of genes that promote cell proliferation (e.g. the cell cycle gene, *Cyclin E*) and inhibit apoptosis [e.g. *Drosophila* inhibitor of apoptosis 1 (*Diap1*)]. Mutations in all the core Hippo pathway components result in a super-competitor phenotype in a clonal context, where the mutant cells out-compete the surrounding wild-type cells (Tyler et al., 2007). Conversely, high levels of Hippo signaling confer a loser cell phenotype. The mechanism by which the Hippo pathway is regulated in different cell competition scenarios has not been precisely discerned. However, in *crb* mutant super-competitor clones Expanded (an upstream negative regulator of Hippo signaling) is mislocalized, which leads to impaired Hippo signaling (Chen et al., 2010; Robinson et al., 2010).

For many years, a question that remained unanswered in the field of cell competition was how are loser and winner cells communicating with each other to discern the fitness status of

[1]Department of Biochemistry and Chemistry, La Trobe Institute for Molecular Sciences, La Trobe University, Melbourne, Victoria 3086, Australia. [2]Blood Cells and Blood Cancer Division, The Walter and Eliza Hall Institute of Medical Research, Melbourne, Victoria 3052, Australia. [3]Department of Medical Biology, University of Melbourne, Melbourne, Victoria 3010, Australia. [4]Genome Engineering and Cancer Modelling Program, Olivia Newton-John Cancer Research Institute, Melbourne, Victoria 3084, Australia.

*Author for correspondence (H.Richardson@latrobe.edu.au)

ⓘ N.F.-L., 0000-0002-9123-9636; M.P., 0000-0001-7898-962X; J.E.L.M., 0000-0001-6442-9947; H.E.R., 0000-0003-3852-4953

neighboring cells and subsequently orchestrate their elimination? A recently discovered mechanism of cell competition involving a direct receptor-ligand interaction is the Sas-Ptp10D axis, which has been shown to particularly concern loser cells with compromised cell polarity within epithelial tissues (Yamamoto et al., 2017). Ptp10D is a receptor-type tyrosine phosphatase that undergoes re-localization to the lateral membrane in polarity-deficient cells, while its ligand Stranded at second (Sas) is similarly re-localized to the lateral membrane in neighboring wild-type cells. Ptp10D activation via Sas binding inhibits the EGFR-Ras signaling pathway, enabling pro-apoptotic JNK signaling and the elimination of the polarity-deficient cells.

Apico-basal cell polarity regulators of the Scribble module – Scrib, Dlg and Lgl – are known to regulate the actin cytoskeleton via various conserved interactions with cytoskeletal regulators in both *Drosophila* and mammalian cells (reviewed by Elsum et al., 2012; Humbert et al., 2015; Mack and Georgiou, 2014). For example, our previous studies have revealed genetic interactions between *Drosophila dlg* and actin cytoskeleton regulator genes, such as *RhoGEF2* and *Rho* (Brumby et al., 2011), and *scrib* mutant cells have been shown to exhibit elevated Myosin II activity (Külshammer and Uhlirova, 2013). Moreover, *Drosophila* Lgl binds to and negatively regulates Myosin II function (Betschinger et al., 2005; Strand et al., 1994). Similar to the loser cell phenotype exhibited by cells lacking Scribble module proteins when adjacent to wild-type cells (Brumby and Richardson, 2003; Grzeschik et al., 2007; Igaki et al., 2006; Menendez et al., 2010), changes in cell morphology governed by elevated expression of actin cytoskeleton regulators (such as RhoGEF2 and Rho) also produce a loser phenotype in a clonal context (Brumby et al., 2011; Khoo et al., 2013). Rho signaling proceeds via RhoGEF2-mediated activation of Rho, which then acts via the effectors Diaphanous, LIM Kinase and Protein Kinase N to regulate F-actin polymerization and the Rok protein kinase, which in turn phosphorylates and activates Myosin II to increase actin-myosin contractility (Barrett et al., 1997; Hacker and Perrimon, 1998; Jaffe and Hall, 2005). Given the link between cell polarity impairment and actin cytoskeleton protein deregulation (Mack and Georgiou, 2014), we sought to investigate whether overexpression of *RhoGEF2* (*RhoGEF2^OE*), which deregulates the actin cytoskeleton through aberrant Rho signaling without substantial alterations in cell polarity (Khoo et al., 2013), could induce cell competition, and whether this process involves Sas-Ptp10D signaling. Therefore, we overexpressed *RhoGEF2* in clones in the *Drosophila melanogaster* eye-antennal epithelium, as previously conducted (Khoo et al., 2013), and examined the effect of Ptp10D depletion. We report that *RhoGEF2^OE* cells undergo cell competition when surrounded by wild-type cells and that Sas-Ptp10D signaling is involved in the elimination of these cells. Mechanistically, in cells with moderate *RhoGEF2* overexpression, the Hippo pathway is activated but the EGFR-Ras and JNK pathways are not affected, and Ptp10D knockdown prevents cell competition by inhibiting the Hippo pathway. Additionally, we show that the apical cell polarity protein Crb also plays a role in the elimination of *RhoGEF2^OE* cytoskeletal deregulated cells via cell competition. In cells with high *RhoGEF2* overexpression, Hippo and JNK signaling were activated, and EGFR-Ras signaling was reduced, and in this scenario *crb* loss normalized these pathways. Finally, we found that *crb* mutant cells exhibited reduced levels of membrane localization of Ptp10D, suggesting that Crb might be important for Ptp10D signaling-mediated elimination of *RhoGEF2^OE* clones. The findings from this study expand the role of Ptp10D signaling in cell competition and demonstrate that

it also plays a role in the elimination of *RhoGEF2^OE* cells, where the actin cytoskeleton is deregulated.

## RESULTS

### Ptp10D is required for *scrib* mutant clone elimination, but Ptp10D overexpression does not enhance cell competition

The discovery in *Drosophila* of the role of the Sas-Ptp10D signaling system in polarity-deficient cell competition marked a significant breakthrough (Yamamoto et al., 2017). However, recent controversy has emerged regarding the obligatory role of Ptp10D in this context, since Gerlach et al. (2022) reported that depleting Ptp10D in *scrib^−/−* clones, using the same *Ptp10D-RNAi* stock as used in the Yamamoto et al. study did not rescue clone size in third-instar (L3) *Drosophila* eye imaginal discs under their laboratory conditions. However, another recent study also utilizing the same *Ptp10D-RNAi* stock as used by Yamamoto et al. found that Ptp10D was necessary for the elimination of polarity disrupted clones (Liu et al., 2022). Consequently, we also attempted to replicate the results Yamamoto et al. using our own stocks and laboratory conditions.

To evaluate this, we performed a clonal knockdown of *Ptp10D* in polarity-disrupted (*scrib^−/−*) cells and measured the relative clone size by quantifying the volume of GFP+ tissue relative to the total volume of the disc (see Materials and Methods). We found that *Ptp10D* knockdown (*Ptp10D^KD*) with *UAS-Ptp10D-RNAi* lines, was able to partially rescue the elimination of *scrib^−/−* clones (compare Fig. 1C and 1D to 1B; quantified in 1E), consistent with the findings of Yamamoto et al. and Liu et al. However, we noted that the increase in clone size was not as pronounced as that reported by the previous studies, and did not reach wild-type clone size, a discrepancy which might potentially be explained by the use of different *Ptp10D^KD* lines in our experiments. However, given that no rescue at all was observed in the Gerlach et al. (2022) study, it is possible that other cell competition mechanisms might be at play under their study conditions.

Notably, we utilized two *Ptp10D^KD* lines obtained from the Vienna *Drosophila* Research Center (VDRC), numbered 1101 and 1102, different to the studies by Yamamoto et al. (2017), Liu et al. (2022) and Gerlach et al. (2022), which used the *Ptp10D-RNAi* 39086 line from the Bloomington *Drosophila* Stock Center (BDSC). The two *Ptp10D^KD* lines we used in this study exhibited very similar knockdown of Ptp10D in *scrib^−/−* L3 larval eye epithelia (~50%) (see Fig. S1, compare Fig. S1D′ and S1E′ to S1F′; quantified in Fig. S1H); however, we found that the *Ptp10D^KD* line 1101 induced a stronger adult eye phenotype than the *Ptp10D^KD* line 1102 in a *scrib* mutant background (compare Fig. 1H and 1I). It is possible that the *Ptp10D^KD* line 1101 might induce a greater reduction of Ptp10D in *scrib^−/−* clones at the pupal-adult stages, accounting for the more severe adult eye phenotype. However, for both *Ptp10D^KD* lines, the *scrib^−/−* *Ptp10D^KD* adult eyes were similar to that observed by Yamamoto et al., showing greater disruptions and necrosis than *scrib^−/−* alone adult eyes, and were contrary to that observed by Gerlach et al., where no necrotic tissue or further disruption to the *scrib^−/−* adult eye structure was observed. Given that the effect of *Ptp10D^KD* line 1101 was stronger, it was used for all subsequent experiments.

Furthermore, in support of the Yamamoto et al. study, we found that *Sas^KD* in surrounding wild-type cells also rescued *scrib^−/−* cell elimination (Fig. S2). Interestingly, Gerlach et al. conducted the same experiment and although they did observe a similar patchy or necrotic adult eye phenotype as observed by us and Yamamoto et al., they did not observe an increase in *scrib^−/−* clone size.

Journal of Cell Science

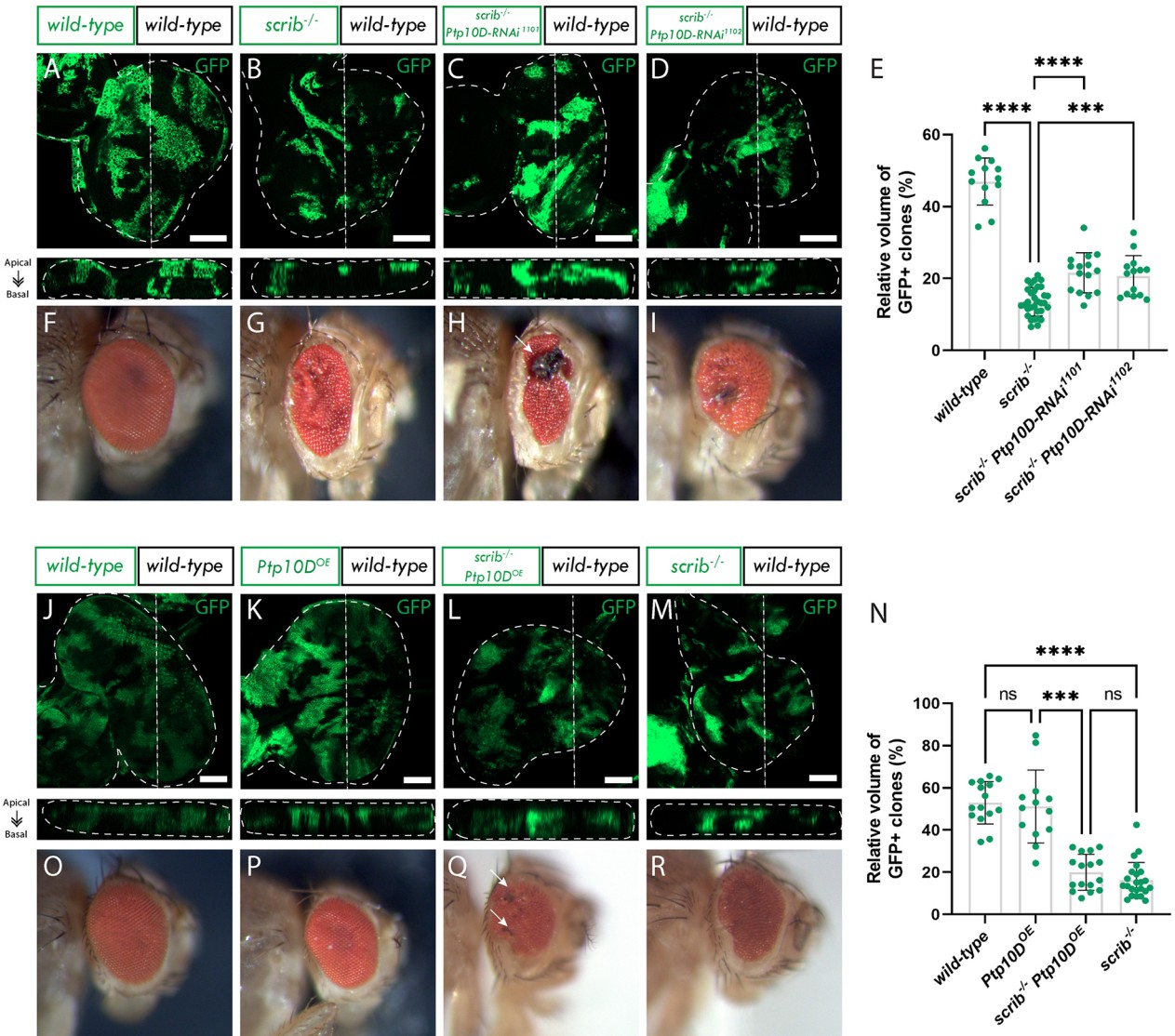

**Fig. 1. Ptp10D is required for *scrib* mutant clone elimination and *Ptp10D* overexpression does not enhance cell competition.** (A–D) Eye discs of *ey-FLP-MARCM*-induced mosaics (clones are marked by the presence of GFP and wild-type tissue is unmarked): (A) wild-type; (B) *scrib⁻/⁻*; (C) *scrib⁻/⁻ Ptp10D-RNAi¹¹⁰¹*; (D) *scrib⁻/⁻ Ptp10D-RNAi¹¹⁰²*. (E) Quantifications of relative GFP+ clone volume [mean±s.d.; wild-type (*n*=13); *scrib⁻/⁻* (*n*=32); *scrib⁻/⁻ Ptp10D-RNAi¹¹⁰¹* (*n*=15); *scrib⁻/⁻ Ptp10D-RNAi¹¹⁰²* (*n*=14)]. ****$P<0.0001$; ***$P=0.0003$ (one-way ANOVA, $P<0.05$, with Tukey's multiple comparison test). (F–I) Adult eyes of corresponding eye imaginal discs to those shown directly above (arrow in H indicate necrotic tissue). (J–M) Eye discs of *ey-FLP-MARCM*-induced mosaics (clones are marked by the presence of GFP and wild-type tissue is unmarked): (J) wild-type; (K) *Ptp10D overexpression (Ptp10Dᴼᴱ)*; (L) *scrib⁻/⁻ Ptp10Dᴼᴱ*; (M) *scrib⁻/⁻*. (N) Quantifications of relative GFP+ clone volume [mean±s.d.; wild-type (*n*=15); *Ptp10Dᴼᴱ* (*n*=13); *scrib⁻/⁻ Ptp10Dᴼᴱ* (*n*=16); *scrib⁻/⁻* (*n*=23). ****$P<0.0001$; ***$P=0.0009$; ns, not significant (Kruskal–Wallis test, $P<0.05$, with Dunn's multiple comparison test). (O–R) Adult eyes of corresponding eye imaginal discs to those shown directly above (arrows in Q indicate necrotic speckles). Below each confocal image is an *xz* cross section of the corresponding eye-antennal disc from the apical (top) to basal (bottom) edge, with the position of the chosen *xz* sections indicated by a vertical dotted line in the *xy* images. Dotted lines surrounding discs illustrate disc boundaries. Scale bars: 20 µm. Adult eye images were all taken at the same magnification.

Additionally, we tested whether the levels of Ptp10D might be rate-limiting for cell competition of polarity-impaired cells – i.e. does increasing the amount of Ptp10D in *scrib⁻/⁻* clones induce stronger cell elimination? We observed that overexpression of *Ptp10D* in *scrib⁻/⁻* clones did not result in any significant further reduction in clone size (compare Fig. 1L and 1M; quantified in 1N). Adult eyes resulting from the *scrib⁻/⁻* clones overexpressing *Ptp10D* (*Ptp10Dᴼᴱ*) were a similar size to *scrib⁻/⁻* mosaics, but some contained small necrotic speckles (compare Fig. 1Q and 1R). Thus, the levels of Ptp10D in *scrib⁻/⁻* mutant clones are sufficient to be fully activated by Sas in the adjacent wild-type cells.

## Cytoskeletal deregulated clones are eliminated by cell competition

Given that cytoskeletal deregulation occurs downstream of cell polarity disruption (Brumby et al., 2011; Elsum et al., 2012), and as Myosin II activation occurs in *scrib* mutant clones (Külshammer and Uhlirova, 2013) as well as *RhoGEF2ᴼᴱ* clones (Khoo et al., 2013), we sought to investigate whether directly deregulating the actin cytoskeleton (without overtly affecting cell polarity) by overexpressing *RhoGEF2ᴼᴱ* also triggered cell competition. Although our previous studies have revealed that *RhoGEF2ᴼᴱ* reduced clone size (Brumby et al., 2011; Khoo et al., 2013), it was not quantified.

As anticipated, expression of $RhoGEF2^{OE}$ in clones resulted in a significant reduction in the volume of $RhoGEF2^{OE}$ GFP⁺ tissue, which constituted only an average of 10% of the total eye disc volume, compared to the 50% average volume of wild-type GFP⁺ tissue in control eye disc mosaics (compare Fig. 2A and 2B; quantified in 2D). Importantly, it was previously observed that when *RhoGEF2* was overexpressed in the whole eye tissue, these cells did not exhibit reduced growth but instead overgrew (Brumby et al., 2011; Khoo et al., 2013), indicating that $RhoGEF2^{OE}$ cells are not intrinsically 'sick', and suggesting it is only in a clonal context that they are eliminated by cell competition. Interestingly $RhoGEF2^{OE}$ clonal tissue presented with a cyst-like phenotype (Fig. 2C). It has previously been described in wing imaginal discs that this rounded ball-like shape is a consequence of recruitment of Actin, Myosin and Moesin to the interface between cells with different fates, which

induces apical constriction and a reduction in lateral contact area between different cells (Bielmeier et al., 2016).

Next, to explore the potential involvement of the Sas-Ptp10D system in the elimination of $RhoGEF2^{OE}$ clones we assessed the localization of Ptp10D and Sas in $RhoGEF2^{OE}$ eye disc mosaics by performing Ptp10D and Sas immunostains. The results revealed a notable shift in the localization of Ptp10D and Sas; no longer confined to the apical membrane but instead being found more basally, either within or surrounding the $RhoGEF2^{OE}$ clones (compare Fig. 2A′ and 2A″ to 2B′ and 2B″; magnified in C′ and C″). Within $RhoGEF2^{OE}$ clones in eye disc mosaics, quantification showed an ~3-fold average increase in Ptp10D abundance (Fig. 2E), and just under a 1.5-fold average increase for Sas abundance (Fig. 2F). Furthermore, DAPI staining of cell nuclei and the membrane-bound mCD8–GFP marker highlighted a concentration of Ptp10D and Sas at the cell membranes (Fig. 2C–C″),

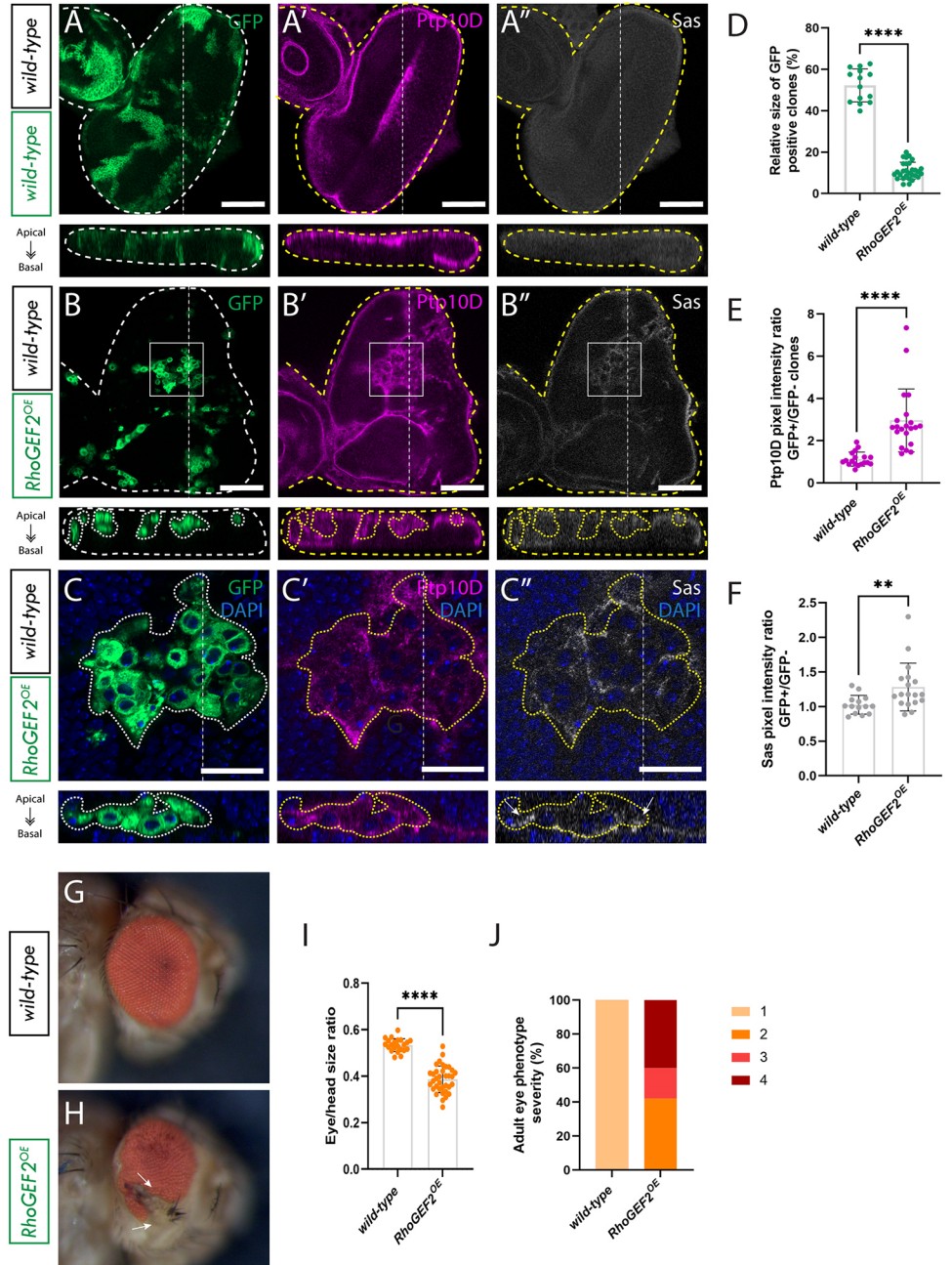

**Fig. 2. Ptp10D and Sas accumulate at the $RhoGEF2^{OE}$ clone cell membrane in $RhoGEF2^{OE}$ eye disc mosaics.** (A) Eye disc of wild-type *ey-FLP-MARCM*-induced mosaics; (A′) Ptp10D immunostain; (A″) Sas immunostain. (B) Eye disc of $RhoGEF2^{OE}$ *ey-FLP-MARCM*-induced mosaics ($RhoGEF2^{OE}$ clones are marked by the presence of GFP and wild-type tissue is unmarked); (B′) Ptp10D immunostain; (B″) Sas immunostain (rectangles indicate magnified sections shown below). (C) Magnified section of eye disc of *ey-FLP-MARCM*-induced mosaics+DAPI; (C′) Ptp10D immunostain+DAPI; (C″) Sas immunostain+DAPI (arrows indicate Sas accumulation). (D) Quantification of relative GFP+ clone volume [mean±s.d.; wild-type ($n$=14); $RhoGEF2^{OE}$ ($n$=29)]. ****$P$<0.0001 (unpaired two-tailed *t*-test, $P$<0.05). (E) Quantification of Ptp10D immunostain pixel intensity ratio between GFP+ and GFP− clones [mean±s.d.; wild-type ($n$=14); $RhoGEF2^{OE}$ ($n$=18)]. ****$P$<0.0001 (Mann–Whitney test, $P$<0.05). (F) Quantification of Sas immunostain pixel intensity ratio between GFP+ and GFP− clones [mean±s.d.; wild-type ($n$=14); $RhoGEF2^{OE}$ ($n$=18)]. **$P$=0.0036 (Mann–Whitney test, $P$<0.05). (G) Wild-type mosaic adult eye; (H) $RhoGEF2^{OE}$ mosaic adult eye (arrows indicate protrusion); (I) relative eye to head size quantification [mean±s.d.; wild-type ($n$=22); $RhoGEF2^{OE}$ ($n$=35)]. ****$P$<0.0001 (unpaired two-tailed *t*-test, $P$<0.05). (J) Percentages of adult eyes with the following phenotypes 1: wild-type, 2: slight rough eye, 3: rougher eye and/or necrotic speckles 4: rougher eye and/or protrusions (wild-type, $n$=22; $RhoGEF2^{OE}$, $n$=33). Below each confocal image is an *xz* cross section of the corresponding eye-antennal disc from the apical (top) to basal (bottom) edge, with the position of the chosen *xz* sections indicated by a vertical dotted line in the *xy* images. Dotted lines surrounding discs or clones illustrate disc/clone boundaries. Note that folding at the edges of the discs can result in Ptp10D and Sas being observed basally in the *xz* sections in these regions, despite the staining still being localized at the apical membrane. Scale bars: 50 μm (A,B); 20 μm (C). Adult eye images were all taken at the same magnification.

further suggesting that Ptp10D and Sas might be playing a role in the elimination of these clones. Moreover, the size of adult eyes resulting from $RhoGEF2^{OE}$ eye-antennal disc mosaics was significantly diminished compared to that in wild-type adult eyes (compare Fig. 2G and 2H; quantified in 2I). This observation implies that $RhoGEF2^{OE}$ clones are being eliminated from the tissue at a higher rate than the surrounding wild-type cells are able to replace them (through compensatory cell proliferation mechanisms), analogous to the situation observed in $scrib^{-/-}$ adult eye mosaics. We then also measured the rugosity or deformity of $RhoGEF2^{OE}$ mosaic adult eye phenotypes on a four-point scale (where a wild-type appearance is 1); 40% exhibited slight tissue rugosity akin to that observed in $scrib^{-/-}$ adult eye mosaics (rated 2), whereas 20% of $RhoGEF2^{OE}$ adult eyes displayed more pronounced rugosity and/or necrotic speckles (rated 3), and 40% exhibited epithelial protrusions, some resembling parts of the antennae (rated 4) (compare Fig. 2G and 2H; plotted in 2J). These data show that cytoskeletal disorganization due to $RhoGEF2^{OE}$ expression markedly impacts eye and antenna structure development. Importantly, our results reveal that $RhoGEF2^{OE}$ clones undergo cell competition and that Ptp10D and Sas protein localization is altered to be more basal in and surrounding these clones.

### Ptp10D knockdown increases $RhoGEF2^{OE}$ clone volume

To investigate if Ptp10D is playing a role in $RhoGEF2^{OE}$ cell competition, we knocked down $Ptp10D$ in $RhoGEF2^{OE}$ clones and measured the volume of the $RhoGEF2^{OE}$ GFP$^+$ tissue. If Ptp10D is indeed a critical mediator of $RhoGEF2^{OE}$ clone elimination, then the downregulation of $Ptp10D$ would be expected to rescue the elimination of $RhoGEF2^{OE}$ clones. To account for $GAL4/UAS$ dosage effects and ensure equivalent expression levels of $RhoGEF2^{OE}$, a control $UAS$ construct ($UAS-Dicer2$) was introduced into the $UAS-RhoGEF2^{OE}$ stock. In this scenario the levels of $RhoGEF2^{OE}$ are reduced to half, which we considered to be 'moderate overexpression'. Analysis of the upregulation of Ptp10D in these clones was as expected less than with 'high overexpression' of $RhoGEF2$ (Fig. 2), but a significant 1.5-fold increase in Ptp10D abundance was still observed (Fig. S3). Concurrently, effective Ptp10D knockdown was confirmed when $Ptp10D^{KD}$ was expressed alone (with a $UAS-myr-RFP$ dosage control) and in combination with $RhoGEF2^{OE}$, resulting in an average 50% reduction in Ptp10D antibody staining intensity (Fig. S3). Quantification of GFP$^+$ tissue volumes revealed that $RhoGEF2^{OE}$ $Dicer2$ clones did not show significant size reduction compared to wild-type eye disc mosaics (compare Fig. 3A and 3B; quantified in 3I), in contrast to the observations with high $RhoGEF2^{OE}$ expression (no $Dicer2$) (Fig. 2). However, $Ptp10D^{KD}$ $RhoGEF2^{OE}$ clones were significantly increased in volume compared with $RhoGEF2^{OE}$ $Dicer2$ clones (compare Fig. 3B and 3C; quantified in 3I), indicating that Ptp10D is playing a role in controlling $RhoGEF2^{OE}$ clone size. Importantly, Ptp10D knockdown (again together with $UAS-myr-RFP$) did not increase the volume of wild-type cells (compare Fig. 3A and 3D; quantified in 3I). Taken together, these results indicate that Ptp10D has no effect in wild-type cells, but functions in $RhoGEF2^{OE}$ clones to limit their growth and/or reduce their competitiveness. Following on from previous findings (Gerlach et al., 2022), we performed this experiment using a different food recipe. We utilized a low-protein food recipe (instead of our regular high yeast, molasses-based food recipe), given that we hypothesized that it might make wild-type cells more competitive relative to cytoskeletal deregulated cells. Interestingly, in low-protein food, $RhoGEF2^{OE}$ $Dicer2$ clones were significantly smaller than in the wild-type controls (Fig. S4), consistent with previous studies showing that environmental conditions are key to cell competition processes (Agrawal et al., 2016; Gerlach et al., 2022;

Sanaki et al., 2020). Furthermore, $Ptp10D$ knockdown in $RhoGEF2^{OE}$ clones, in this setting, rescued the reduced clone size (Fig. S4), indicating that Ptp10D is involved in cell competition of $RhoGEF2^{OE}$ clones. As we were interested in the clonal size increase of the $RhoGEF2^{OE}$ $Ptp10D^{KD}$ clones compared to $RhoGEF2^{OE}$ $Dicer2$, we continued to utilize our regular high yeast, molasses-based food for subsequent experiments.

Next, we explored cell death in $RhoGEF2^{OE}$ $Dicer2$ and $RhoGEF2^{OE}$ $Ptp10D^{KD}$ clones. Accordingly, we conducted an immunostain for the active (cleaved) form of the apoptotic protein Caspase-3, a marker for cells undergoing apoptosis. We measured the spot density ratio of cleaved Caspase-3-positive cells between GFP+ and GFP− cells, which normalizes the number of positively marked cells to the tissue volume. A significant increase in cleaved Caspase-3 spot density was observed in $RhoGEF2^{OE}$ $Dicer2$ clones compared to wild-type cells (compare Fig. 3A,A' and B,B'; quantified in 3J). This result was somewhat unexpected, as no significant decrease in the total volume of $RhoGEF2^{OE}$ $Dicer2$ clones (in our regular food) was observed compared to in the wild type. This discrepancy suggests that although these clones undergo cell competition-induced cell death, compensatory cell proliferation mechanisms might be at play in $RhoGEF2^{OE}$ $Dicer2$ tissue to offset any clonal size reduction. When $Ptp10D$ was knocked down in $RhoGEF2^{OE}$ clones, no significant decrease in cleaved Caspase-3 spot density ratio was observed compared to $RhoGEF2^{OE}$ $Dicer2$ clones, but a downward trend was noted (compare Fig. 3B,B' and Fig. 3C,C'; quantified in 3J). However, cleaved Caspase-3 levels in $RhoGEF2^{OE}$ $Ptp10D^{KD}$ clones compared to the wild-type clones were not statistically different (compare Fig. 3A,A' and C,C'; quantified in 3J), suggesting that $RhoGEF2^{OE}$ $Ptp10D^{KD}$ clones undergo less cell death than $RhoGEF2^{OE}$. Additionally, $Ptp10D^{KD}$ clones showed no differences in cleaved Caspase-3 spot density compared to that seen in the wild type, indicating that Ptp10D knockdown alone does not affect cell death (compare Fig. 3A,A' and D,D'; quantified in 3J).

The size of adult eyes was significantly reduced in both $RhoGEF2^{OE}$ $Dicer2$ (Fig. 3F) and $RhoGEF2^{OE}$ $Ptp10D^{KD}$ (Fig. 3G), with no significant differences between them, although each were significantly different to the wild type (Fig. 3E; quantified in 3K). However, a higher percentage of $RhoGEF2^{OE}$ $Dicer2$ adult eyes exhibited deeper folds and more severe rugosities than $RhoGEF2^{OE}$ $Ptp10D^{KD}$ adult eyes (Fig. 3E–H, quantified in 3L), suggesting that $Ptp10D$ knockdown partially rescues the severe eye phenotype of $RhoGEF2^{OE}$ mosaics – the opposite of what occurs in the polarity-deficient $Ptp10D^{KD}$ scenario, where $Ptp10D^{KD}$ increases the severity of the adult eye phenotype (compare Fig. 1H and 3G). In summary, these findings indicate that Ptp10D plays a role in limiting the growth of (or reducing the competitiveness of) $RhoGEF2^{OE}$ clones, as $Ptp10D^{KD}$ leads to increased $RhoGEF2^{OE}$ clone size and partially rescues the adult eye phenotype.

### $Ptp10D$ knockdown reduces Hippo signaling in $RhoGEF2^{OE}$ clones

Next, we investigated the impact of Ptp10D knockdown on the Hippo pathway in $RhoGEF2^{OE}$ clones by staining for the Hippo pathway target Diap1 (Yoo et al., 2002). As expected, Diap1 was found to be upregulated in $RhoGEF2^{OE}$ $Ptp10D^{KD}$ clones compared to $RhoGEF2^{OE}$ $Dicer2$ clones (compare Fig. 4B″ and F'G' to 4C' and 4G'; quantified in 4I), indicating impaired Hippo pathway signaling. No differences in Diap1 levels were observed between $RhoGEF2^{OE}$ $Dicer2$ and the wild-type control (Fig. 4A',E') or $Ptp10D-RNAi$ alone (Fig. 4D',H').

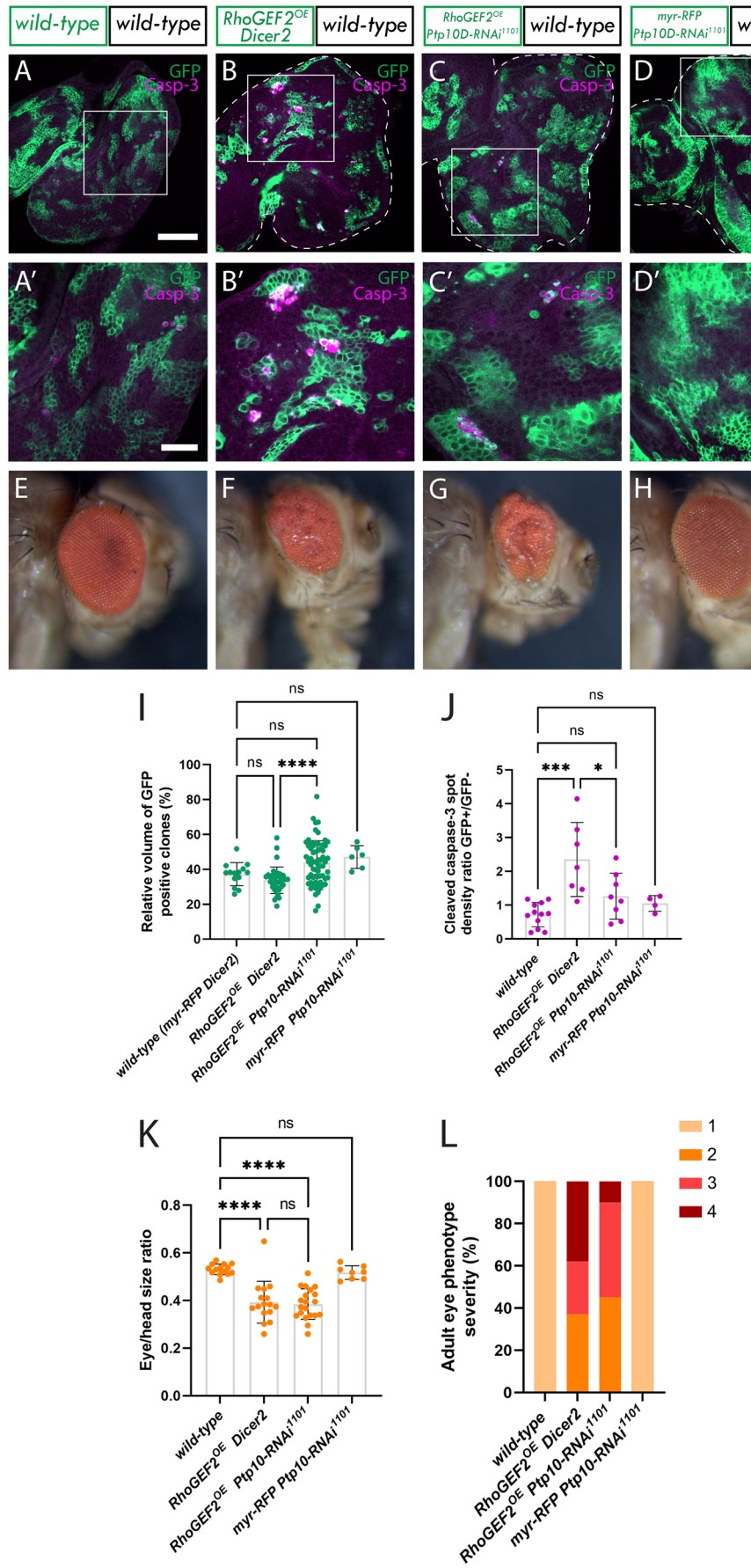

**Fig. 3. *Ptp10D* knockdown increases *RhoGEF2^{OE}* clonal growth in *RhoGEF2^{OE}* eye disc mosaics.** (A–D) Eye discs of *ey-FLP-MARCM*-induced mosaics (clones are marked by the presence of GFP and wild-type tissue is unmarked): (A) wild-type; (B) *RhoGEF2^{OE} Dicer2*; (C) *RhoGEF2^{OE} Ptp10D-RNAi* (VDRC 1102 line); (D) *myr-RFP Ptp10D-RNAi* (VDRC 1101 line), showing cleaved Caspase-3 immunostains, which is an indicator of cell death by apoptosis (white squares indicate magnified sections shown below). (A′–D′) Magnified sections of eye discs of *ey-FLP-MARCM*-induced mosaics showing cleaved caspase-3 immunostains. (E–H) Adult eyes of directly above corresponding eye imaginal discs. (I) Quantification of the relative GFP+ clone volume [mean±s.d.; wild-type (*n*=14); *RhoGEF2^{OE} Dicer2* (*n*=36); *RhoGEF2^{OE} Ptp10D-RNAi* 1101 (*n*=65); *myr-RFP Ptp10D-RNAi* 1101 (*n*=5)]. ****$P$<0.0001; ns, not significant (one-way ANOVA, $P$<0.05, with Tukey's multiple comparison test). (J) Cleaved Caspase-3 spot density ratio, which normalizes the amount of cleaved Caspase-3 positive cells to the total amount of GFP tissue [mean±s.d.; wild-type (*n*=13); *RhoGEF2^{OE} Dicer2* (*n*=7); *RhoGEF2^{OE} Ptp10D-RNAi* 1101 (*n*=8); *myr-RFP Ptp10D-RNAi* 1101 (*n*=5)]. *$P$=0.0278; ***$P$=0.0004; ns, not significant (Kruskal–Wallis test, $P$<0.05, with Dunn's multiple comparison test). (K) Relative eye to head size quantification [mean±s.d.; wild-type (*n*=14); *RhoGEF2^{OE} Dicer2* (*n*=16); *RhoGEF2^{OE} Ptp10D-RNAi* 1101 (*n*=19); *myr-RFP Ptp10D-RNAi* 1101 (*n*=8)]. ****$P$<0.0001; ns, not significant (Kruskal–Wallis test, $P$<0.05, with Dunn's multiple comparison test). (L) Percentages of adult eyes for each genotype that show 1: normal phenotype, 2: slight rough eye, 3: rougher eye and/or necrotic speckles, 4: rougher eye and/or protrusions [wild-type (*n*=122); *RhoGEF2^{OE} Dicer2* (*n*=16); *RhoGEF2^{OE} Ptp10D^{KD}* (*n*=21); *myr-RFP Ptp10D^{KD}* (*n*=12)]. Dotted lines surrounding discs illustrate disc boundaries. Scale bars: 50 μm (A–D); 20 μm (A′–D′). Adult eye images were all taken at the same magnification.

In the canonical model, Hippo pathway inactivation in $scrib^{-/-}$ $Ptp10D^{KD}$ clones is a consequence of an F-actin accumulation, which depends on the activation of JNK and EGFR-Ras signaling pathways (Yamamoto et al., 2017). Therefore, to further understand how Hippo is being inactivated in our model, we investigated these pathways. Interestingly, we found that whereas F-actin accumulated in moderate $RhoGEF2^{OE}$ clones, it was not significantly altered upon $Ptp10D$ knockdown in moderate $RhoGEF2^{OE}$ clones (Fig. S5), indicating that Hippo inactivation is not triggered by the hyper-accumulation of F-actin. Next, we investigated the JNK signaling pathway, which plays a pivotal role in many types of cell competition (La Marca and Richardson, 2020; Pinal et al., 2019) and is upregulated when $RhoGEF2$ is highly overexpressed (without $UAS$ dosage control) (Khoo et al., 2013). However, in moderate $RhoGEF2^{OE}$ clones (with $Dicer2$), we observed no significant increase of the JNK signaling pathway target Mmp1 (Fig. S6). Given that no JNK pathway upregulation was observed in this setting, we did not investigate whether $Ptp10D^{KD}RhoGEF2^{OE}$ clones affected this signaling pathway, concluding that JNK

pathway modulation was unlikely to be a core driver of the increased size of $RhoGEF2^{OE}$ $Ptp10D^{KD}$clones. Furthermore, via immunostaining, we did not observe any significant changes in the levels of the phosphorylated (active) form of ERK [pERK; also known as Rolled (rl) or MAPK], the canonical downstream effector of Ras signaling. In fact, pERK levels in $RhoGEF2^{OE}$ $Ptp10D^{KD}$ clones were significantly slightly decreased compared to those in the wild-type control (Fig. S7), showing that Ptp10D does not inhibit EGFR-Ras signaling in this context. Altogether, these results suggest that the Hippo pathway is being inactivated via an unknown mechanism in $RhoGEF2^{OE}$ $Ptp10D^{KD}$ clones, that where JNK signaling is not hyperactive, EGFR-Ras signaling is not activated, and that F-actin hyperaccumulation is not present.

## *crb* mutation partially rescues *RhoGEF2^{OE}* clone elimination in eye disc mosaics

As the apical cell polarity protein Crumbs (Crb) is involved in Hippo pathway regulation and in cell competition (Hafezi et al., 2012; Robinson et al., 2010), we wanted to determine whether *crb*

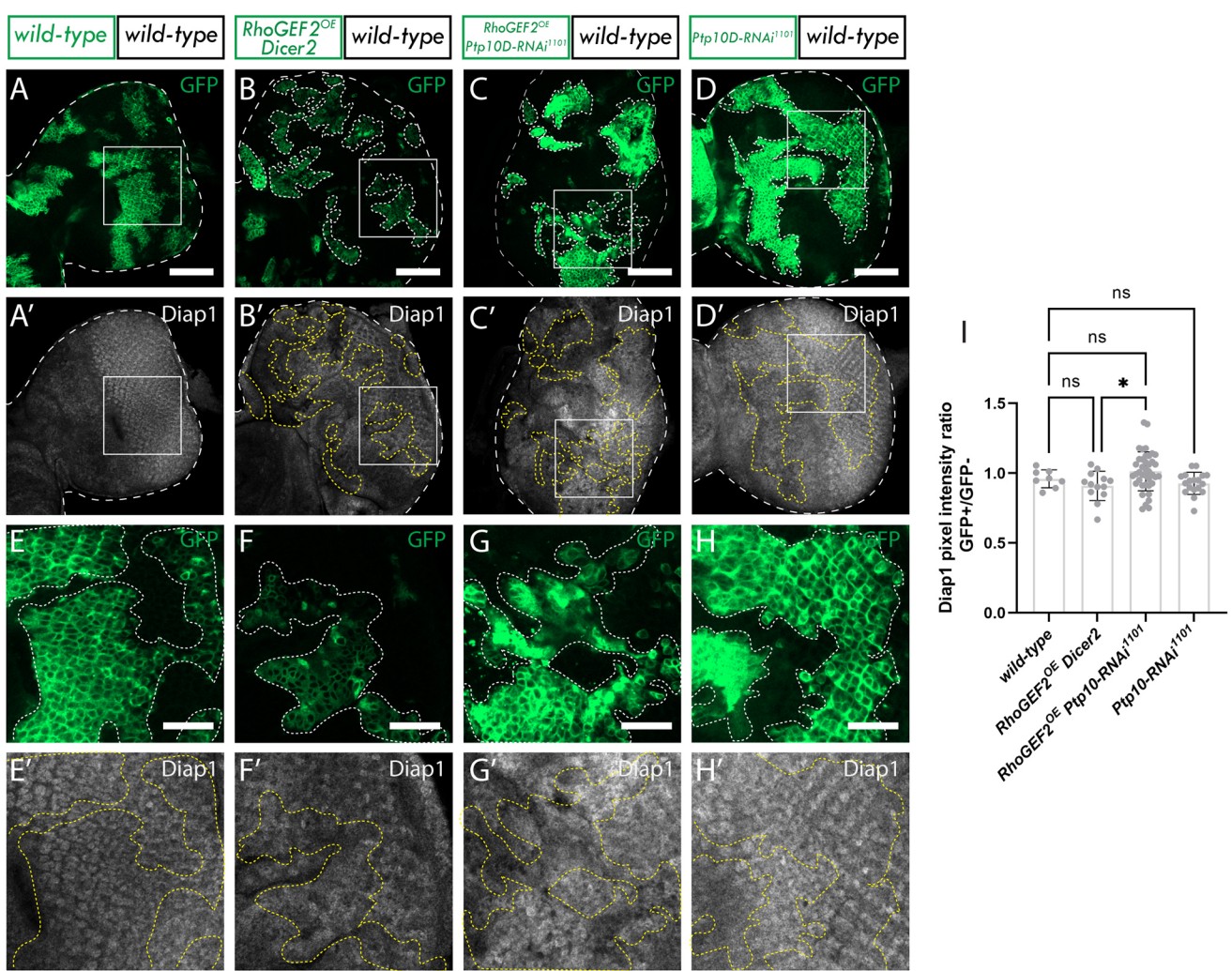

**Fig. 4. *Ptp10D* knockdown reduces Hippo signaling in *RhoGEF2^{OE}* clones.** (A–D) Eye discs of *ey-FLP-MARCM*-induced mosaics (clones are marked by the presence of GFP and *wild-type* tissue is unmarked): (A) wild-type; (B) *RhoGEF2^{OE} Dicer-2*; (C) *RhoGEF2^{OE} Ptp10D-RNAi* (VDRC 1102 line); (D) *myr-RFP Ptp10D-RNAi* (VDRC 1101 line); (A′–D′) Diap1 immunostains from A–D; white squares indicate magnified sections shown below. (E–H) Magnified sections of eye discs from A–D. (E′–H′) Diap1 immunostains from E–H. (I) Quantification of Diap1 pixel intensity ratio between GFP+ and GFP− clones [mean±s.d.; wild-type (*n*=8); *RhoGEF2^{OE} Dicer2* (*n*=13); *RhoGEF2^{OE} Ptp10D-RNAi* 1101 (*n*=40); *myr-RFP Ptp10D-RNAi* 1101 (*n*=18)]. *P*=0.0323; ns, not significant (one-way ANOVA, *P*<0.05, with Tukey's multiple comparison test). Dotted lines surrounding discs or clones illustrate disc/clone boundaries. Scale bars: 50 µm (A–D); 20 µm (E–H).

Journal of Cell Science

was playing a role in cell competition in *RhoGEF2^OE* clones. As shown in Fig. 2, high *RhoGEF2^OE* clones are reduced in GFP+ tissue volume (~10%) compared to in wild-type clones (~50%). However, when *crb* was depleted (using the null allele *crb^{11A22}*) in *RhoGEF2^OE* clones, the average GFP+ tissue volume increased to ~23% (compare Fig. 5B and 5C; quantified in 5I), indicating a partial rescue of the *RhoGEF2^OE* clone elimination phenotype. Additionally, comparing *crb^{−/−}* clones and wild-type clones revealed a slight, although non-significant, increase in clone volume (compare Fig. 5A and 5D; quantified in 5I), in contrast with what was seen in previous reports (Hafezi et al., 2012). The sizes of adult eyes resulting from *RhoGEF2^OE* and *RhoGEF2^OE crb^{−/−}* mosaics were not significantly different to one another but were each significantly decreased compared to the size for wild-type adult eyes, indicating that *crb* loss does not rescue the adult eye size decrease in *RhoGEF2^OE* mosaics (compare Fig. 5E,F,G; quantified in 5K).

However, *crb^{−/−}* mosaic adult eyes were significantly larger than wild-type adult eyes, suggesting that although we did not observe that *crb^{−/−}* clones were significantly larger than wild-type clones at the L3 larval stage they overgrow later in development (compare Fig. 5E and 5H; quantified in 5K).

*RhoGEF2^OE* mosaic adult eyes displayed epithelial protrusions in ~40% of cases (Fig. 5F; quantified in 5L), possibly due to the deregulated actin cytoskeleton. Intriguingly, similar to observations from *RhoGEF2^OE Ptp10D^{KD}* clones, the severity of the adult eye phenotypes in *RhoGEF2^OE crb^{−/−}* mosaics was reduced, with 80% showing only mild rugosities (rated 2) and no protrusions being observed in the remaining 20% (Fig. 5G; quantified in 5L). Taken together, these data suggest that Crb is necessary for the elimination of *RhoGEF2^OE* clones and that although *crb* loss in *RhoGEF2^OE* clones does not impact adult eye size, it modulates the severity of the *RhoGEF2^OE* adult eye phenotype.

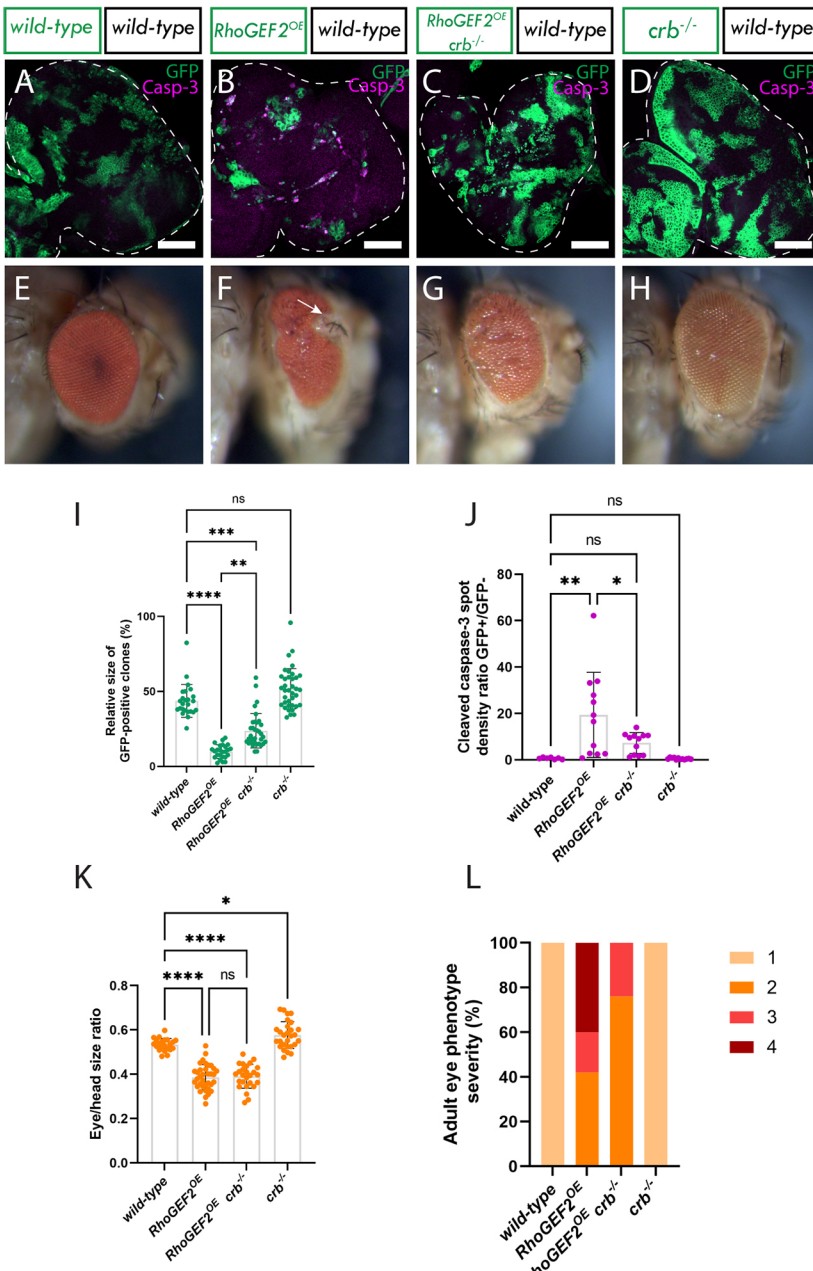

**Fig. 5. *crb* mutation results in a partial rescue of *RhoGEF2^OE* clone elimination in eye disc mosaics.** (A–D) Eye discs of *ey-FLP-MARCM*-induced mosaics (clones are marked by the presence of GFP and wild-type tissue is unmarked): (A) wild-type; (B) *RhoGEF2^OE*; (C) *RhoGEF2^OE crb^{−/−}*; (D) *crb^{−/−}*, showing cleaved Caspase-3 immunostains, which is an indicator of cell death by apoptosis. (E–H) Adult eyes of directly above corresponding eye imaginal discs. (I) Quantification of the relative GFP+ clone volume [mean±s.d.; wild-type (*n*=25); *RhoGEF2^OE* (*n*=25); *RhoGEF2^OE crb^{−/−}* (*n*=34); *crb^{−/−}* (*n*=37)]. **P=0.081; ***P=0.0009; ****P<0.0001; ns, not significant (Kruskal–Wallis test, *P*<0.05, with Dunn's multiple comparison test). (J) Cleaved Caspase-3 spot density ratio, which normalizes the amount of cleaved Caspase-3-positive cells to the total amount of GFP tissue [mean±s.d.; wild-type (*n*=7); *RhoGEF2^OE* (*n*=12); *RhoGEF2^OE crb^{−/−}* (*n*=12); *crb^{−/−}* (*n*=12)]. *P=0.0261; **P=0.0019; ns, not significant (one-way ANOVA, *P*<0.05, with Tukey's multiple comparison test). (K) Relative eye to head size quantification [mean±s.d.; wild-type (*n*=22); *RhoGEF2^OE* (*n*=35); *RhoGEF2^OE crb^{−/−}* (*n*=26); *crb^{−/−}* (*n*=28)]. *P=0.0269; ****P<0.0001; ns, not significant (one-way ANOVA, *P*<0.05, with Tukey's multiple comparison test). (L) Percentages of adult eyes for each genotype that show 1: normal phenotype, 2: slight rough eye, 3: rougher eye and/or necrotic speckles, 4: rougher eye and/or protrusions [wild-type (*n*=22); *RhoGEF2^OE* (*n*=33); *RhoGEF2^OE crb^{−/−}* (*n*=20); *crb^{−/−}* (*n*=10)]. Dotted lines surrounding discs illustrate disc boundaries. Scale bars: 50 µm. Adult eye images were all taken at the same magnification.

To investigate whether the loss of *crb* reduced cell death in *RhoGEF2*$^{OE}$ clones, we conducted immunostaining for cleaved Caspase-3. The spot density ratio of cleaved caspase-3-positive cells between GFP+ and GFP− cells was measured and, as anticipated, we observed a significantly higher spot density ratio in GFP+ clones in *RhoGEF2*$^{OE}$ clones alone compared to in wild-type (compare Fig. 5A and 5B; quantified in 5J), indicating increased cell death. Conversely, *crb*$^{−/−}$ *RhoGEF2*$^{OE}$ clones exhibited significantly fewer cleaved Caspase-3-positive cells, towards wild-type levels (compare Fig. 5B and 5C; quantified in 5J), indicating that Crb is involved in cell death of *RhoGEF2*$^{OE}$ clones through cell competition. Moreover, *crb*$^{−/−}$ clones alone had a similar number of cleaved Caspase-3-positive cells to the wild-type control (compare Fig. 5A and 5D; quantified in 5J), consistent with their resistance to elimination by cell competition.

### *crb* loss decreases apical F-actin accumulation, partially rescues the elevated JNK signaling, rescues EGFR-Ras signaling downregulation and reduces the elevated Hippo signaling in *RhoGEF2*$^{OE}$ clones

To investigate whether the loss of *crb* in *RhoGEF2*$^{OE}$ clones is inducing cell death by inhibiting the JNK signaling pathway, we performed immunostaining for the JNK pathway target Mmp1. As expected, we found that Mmp1 was elevated in these high *RhoGEF2*$^{OE}$ clones relative to what was seen in wild-type control clones [Fig. 6A′ (and control in Fig. S8)] consistent with previous work (Khoo et al., 2013). Upon *crb* loss in *RhoGEF2*$^{OE}$ clones, Mmp1 levels showed a significant reduction compared to *RhoGEF2*$^{OE}$ alone, towards wild-type levels (compare Fig. 6A′ and 6B′; quantified in 6C). These results suggest that Crb contributes to the upregulation of JNK signaling in *RhoGEF2*$^{OE}$ clones. We then assessed the involvement of the EGFR-Ras signaling pathway in the growth of *RhoGEF2*$^{OE}$ *crb*$^{−/−}$ clones by immunostaining for pERK. High *RhoGEF2*$^{OE}$ clones led to pERK levels being diminished compared to wild type [Fig. 6D and D‴ (controls in Fig. S8); quantified in 6F], and upon *crb* loss in *RhoGEF2*$^{OE}$ clones, pERK levels were rescued to wild-type levels (compare Fig. 6D′ and 6D‴ to 6E′ and 6E‴; quantified in 6F), indicating that Crb plays a role in inhibiting EGFR-Ras signaling in *RhoGEF2*$^{OE}$ clones. Next, we tested whether Crb was involved in F-actin accumulation in *RhoGEF2*$^{OE}$ clones. Utilizing phalloidin staining, we confirmed the previously reported observation that F-Actin accumulates in *RhoGEF2*$^{OE}$ clones (Khoo et al., 2013), and furthermore that, upon loss of *crb*, there was a complete rescue of F-actin accumulation (compare Fig. 6G′ and 6H′; quantified in 6I). Additionally, rescue of the cyst-like phenotype of *RhoGEF2*$^{OE}$ clones (compare Fig. 6C and 6B; a more detailed cyst-like phenotype is shown in Fig. 2B and C) was observed upon *crb* loss. Taken together, these findings show that *crb* loss in *RhoGEF2*$^{OE}$ clones reduces F-actin accumulation and cyst formation.

Importantly, Crb serves as a Hippo pathway regulator, as mutations in *crb* lead to inhibition of the Hippo pathway and the expression of genes that facilitate cell proliferation and inhibit cell death (Robinson et al., 2010). To investigate the impact of *crb* loss on the Hippo pathway in *RhoGEF2*$^{OE}$ clones, we analyzed expression of the Hippo pathway target Diap1. In *RhoGEF2*$^{OE}$ clones alone, Diap1 levels were notably reduced compared to wild-type clones [Fig. 6J′ and J‴ (controls in Fig. S8); quantified in Fig. 6L], consistent with increased Hippo signaling in *RhoGEF2*$^{OE}$ clones. Conversely, and as expected, *RhoGEF2*$^{OE}$ *crb*$^{−/−}$ clones exhibited increased Diap1 staining compared with *RhoGEF2*$^{OE}$ clones, indicating decreased Hippo pathway activity (compare

Fig. 6J′ and 6J‴ to Fig. 6K′ and 6K‴; quantified in 6L). Interestingly, in our experiments *crb*$^{−/−}$ clones did not elevate Diap1 expression relative to wild-type (Fig. S8; quantified in Fig. 6L), suggesting that in *RhoGEF2*$^{OE}$ clones *crb* loss is not simply increasing Diap1 levels through a counteractive effect. Thus, Crb might be playing a specific role in *RhoGEF2*$^{OE}$ cell competition by inducing Hippo pathway activity. Altogether, these results show that higher expression of *RhoGEF2*$^{OE}$ results in elevated JNK signaling, reduced EGFR-Ras signaling and elevated Hippo signaling [similar to what occurs in *scrib* mutant clones (Yamamoto et al., 2017)], and that *crb* loss rescues these signaling pathway defects.

### Ptp10D is downregulated in *crb* mutant clones

Finally, we explored whether Crb might be required for Ptp10D protein abundance or membrane localization, since it has been previously reported that it aids in plasma membrane localization of other proteins (Fletcher et al., 2012; Hafezi et al., 2012). We investigated the effect of *crb* loss on Ptp10D protein levels and localization by conducting confocal Z-section analysis of Ptp10D immunostaining in *crb*$^{−/−}$ mosaic eye discs. This examination revealed a notable decrease in Ptp10D levels within *crb*$^{−/−}$ eye disc clones, particularly at the apical membrane where it typically localizes (compare Fig. 7A and 7B; quantified in 7C). This result suggests a potential role for Crb in regulating the protein abundance or localization of Ptp10D at the apical membrane. Furthermore, this finding poses an important question of whether Crb is involved in promoting Sas-Ptp10D-mediated cell competition through its effect on Ptp10D abundance at the apical membrane.

## DISCUSSION

In this manuscript, we first explored and expanded upon the findings of previous studies regarding the role of the Ptp10D-Sas axis in polarity-deficient cell elimination (Gerlach et al., 2022; Igaki et al., 2002; Liu et al., 2022; Yamamoto et al., 2017), around which there had been some contention (Gerlach et al., 2022). We also recapitulated observations regarding *RhoGEF2*$^{OE}$ clone elimination by cell competition (Brumby et al., 2011; Khoo et al., 2013), and have further demonstrated a role for Sas-Ptp10D signaling in restricting the growth of clones of these actin cytoskeleton deregulated cells, indicating that loss of polarity is not a prerequisite for inducing Sas-Ptp10D activity. We observed that in cells with moderate *RhoGEF2*$^{OE}$, Ptp10D induced cell competition by activating the Hippo pathway. Additionally, we found that the apical cell polarity protein Crb contributed to the elimination of high *RhoGEF2*$^{OE}$ cells in a clonal context, via a mechanism involving normalization of the elevated JNK and Hippo signaling and the reduced EGFR-Ras signaling in high *RhoGEF2*$^{OE}$ clones. The results of our findings on signaling pathways are summarized in Fig. 8. Furthermore, we have demonstrated that *crb* mutant cells reduce the levels of apical membrane-localized Ptp10D, suggesting that Crb plays a role in Ptp10D signaling by regulating its levels or membrane localization during *RhoGEF2*$^{OE}$ clone elimination. Overall, this study broadens our understanding of Ptp10D signaling in cell competition by revealing its involvement in eliminating cells overexpressing *RhoGEF2*, which exhibit a deregulated actin cytoskeleton.

### The role of Ptp10D in moderate *RhoGEF2*$^{OE}$ clones is Hippo dependent but JNK and EGFR independent

In our moderate *RhoGEF2*$^{OE}$ model, we showed that *Ptp10D*$^{KD}$ resulted in Hippo pathway inhibition, in line with the canonical *scrib*$^{−/−}$ model (Yamamoto et al., 2017). Nevertheless, how the Hippo pathway is inactivated in these setting seems to differ, as Ras

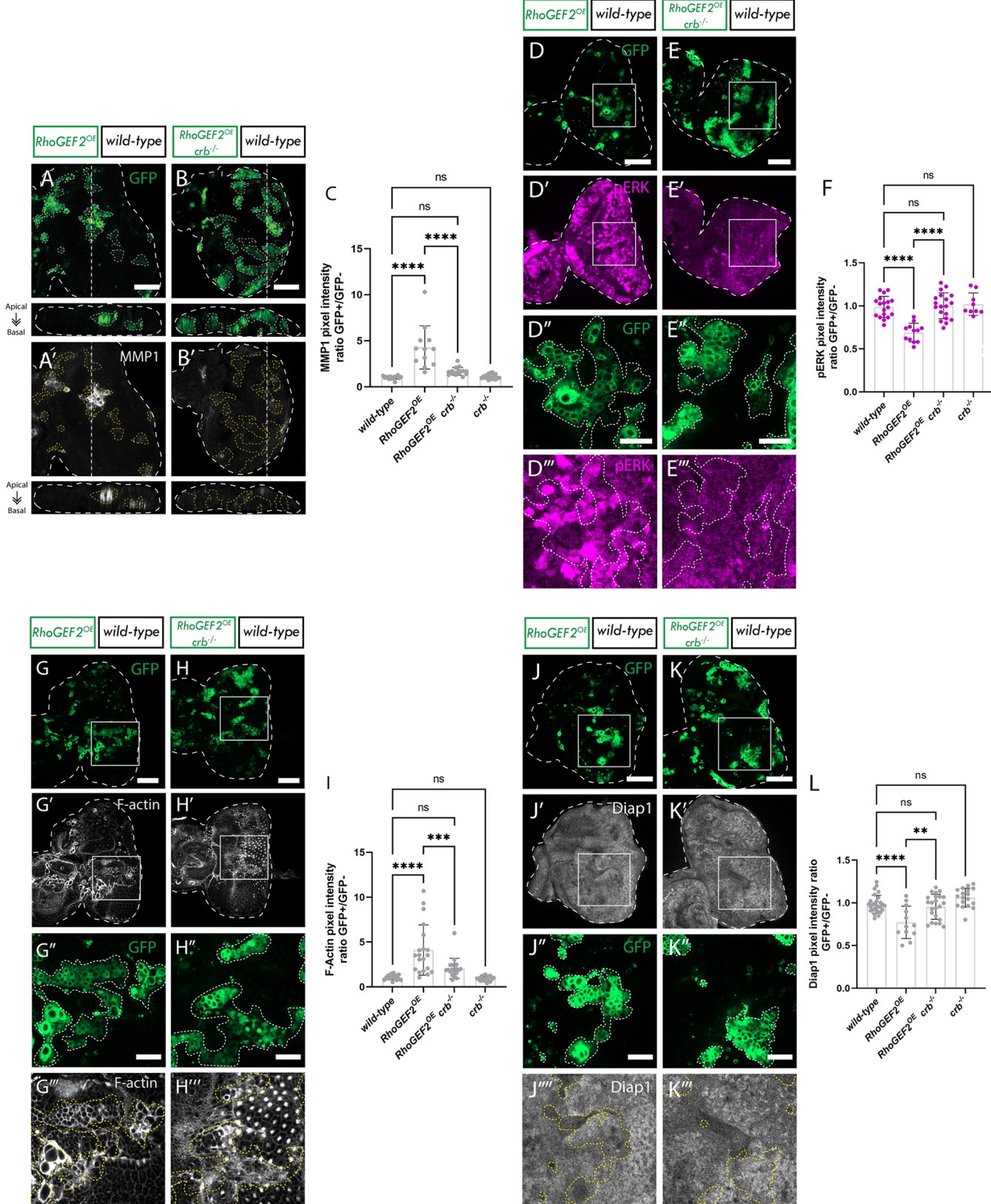

**Fig. 6.** See next page for legend.

signaling was not observed to be reduced in moderate *RhoGEF2^OE^* clones or elevated in *RhoGEF2^OE^ Ptp10D^KD^* clones. Given that Sas-Ptp10D signaling is known to inhibit EGFR-Ras signaling in *scrib* mutant cells (Yamamoto et al., 2017), it is perplexing why moderate *RhoGEF2^OE^* clones did not also lead to reduced Ras signaling. It is possible that moderate *RhoGEF2^OE^* clones do not

**Fig. 6. *crb* loss decreases apical F-Actin accumulation, rescues the elevated JNK signaling, rescues Ras signaling downregulation and reduces the elevated Hippo signaling in *RhoGEF2*$^{OE}$ clones.** (A,B) Eye discs of *ey-FLP-MARCM*-induced mosaics (clones are marked by the presence of GFP and wild-type tissue is unmarked): (A) *RhoGEF2*$^{OE}$; (B) *RhoGEF2*$^{OE}$ *crb*$^{-/-}$; (A′,B′) Mmp1 immunostains from A,B. (C) Quantification of Mmp1 pixel intensity ratio between GFP+ and GFP− clones [mean±s.d.; wild-type (*n*=18); *RhoGEF2*$^{OE}$ (*n*=12); *RhoGEF2*$^{OE}$ *crb*$^{-/-}$ (*n*=15); *crb*$^{-/-}$ (*n*=18)]. ****P<0.0001; ns, not significant (one-way ANOVA, *P*<0.05, with Tukey's multiple comparison test). (D,E) Eye discs of *ey-FLP-MARCM*-induced mosaics (clones are marked by the presence of GFP and wild-type tissue is unmarked): (D) *RhoGEF2*$^{OE}$; (E) *RhoGEF2*$^{OE}$ *crb*$^{-/-}$; (D′,E′) pERK immunostains from D,E; white squares indicate magnified sections shown in panels below; (D″,E″) magnified panels of D,E; (D‴,E‴) pERK immunostains from D″,E″. (F) Quantification of pERK pixel intensity ratio between GFP+ and GFP− clones [mean±s.d.; wild-type (*n*=18); *RhoGEF2*$^{OE}$ (*n*=12); *RhoGEF2*$^{OE}$ *crb*$^{-/-}$ (*n*=19); *crb*$^{-/-}$ (*n*=9)]. ****P<0.0001; ns, not significant (one-way ANOVA, *P*<0.05, with Tukey's multiple comparison test). (G,H) Eye discs of *ey-FLP-MARCM*-induced mosaics (clones are marked by the presence of GFP and wild-type tissue is unmarked): (G) *RhoGEF2*$^{OE}$; (H) *RhoGEF2*$^{OE}$ *crb*$^{-/-}$; (G′,H′) F-actin stains; (G″,H″) magnified panels of G,H; (G‴,H‴) F-actin marker from G″,H″. (I) Quantification of F-Actin pixel intensity ratio between GFP+ and GFP− clones [mean±s.d.; wild-type (*n*=18); *RhoGEF2*$^{OE}$ (*n*=20); *RhoGEF2*$^{OE}$ *crb*$^{-/-}$ (*n*=18); *crb*$^{-/-}$ (*n*=18)]. ***P<0.0008; ****P<0.0001; ns, not significant (one-way ANOVA, *P*<0.05, with Tukey's multiple comparison test). (J,K) Eye discs of *ey-FLP-MARCM*-induced mosaics (clones are marked by the presence of GFP and wild-type tissue is unmarked): (J) *RhoGEF2*$^{OE}$; (K) *RhoGEF2*$^{OE}$ *crb*$^{-/-}$; (J′,K′) Diap1 immunostains from J,K; white squares indicate magnified sections shown in panels below; (J″,K″) magnified panels of J,K; (J‴,K‴) Diap1 stains from J″,K″. (L) Quantification of pERK pixel intensity ratio between GFP+ and GFP− clones [mean±s.d.; wild-type (*n*=28); *RhoGEF2*$^{OE}$ (*n*=12); *RhoGEF2*$^{OE}$ *crb*$^{-/-}$ (*n*=23); *crb*$^{-/-}$ (*n*=18)]. **P=0.0013; ****P<0.0001; ns, not significant (one-way ANOVA, *P*<0.05, with Tukey's multiple comparison test). Controls have been left out of images for simplicity but can be found in Fig. S8. Dotted lines surrounding discs or clones illustrate disc/clone boundaries. In A,A′,B,B′,G,G′,H and H′, images below show an *xz* cross section of the corresponding eye-antennal disc from the apical (top) to basal (bottom) edge, with the position of the chosen *xz* sections indicated by a vertical dotted line in the *xy* images. Scale bars: 50 µm (A,B,D,E,G,H,J,K); 20 µm (D″,E″,J″,K″).

strongly activate the Sas-Ptp10D system, or that the potential inhibition of EGFR-Ras signaling by Sas-Ptp10D in this setting might be counteracted by unknown factors in the moderate *RhoGEF2*$^{OE}$ clones. Elevated JNK signaling was also not detectable in our moderate *RhoGEF2*$^{OE}$ model, in contrast to what occurs in *scrib* mutant clones. F-actin levels are upregulated in moderate *RhoGEF2*$^{OE}$ clones but are not significantly affected by *Ptp10D*$^{KD}$ in these clones. These findings indicate that in these moderate *RhoGEF2*$^{OE}$ *Ptp10D*$^{KD}$ clones, Hippo signaling is being downregulated via a different mechanism to that previously described for *scrib*$^{-/-}$ *Ptp10D*$^{KD}$ clones, where EGFR-Ras and JNK signaling synergize to promote F-actin hyperaccumulation and Hippo pathway inhibition (Yamamoto et al., 2017). Given that *scrib* mutant cells have been shown to possess impaired Hippo pathway signaling dependent on activation of Myosin II (Külshammer and Uhlirova, 2013), and *RhoGEF2*$^{OE}$ cells also activate Myosin II (Khoo et al., 2013), this mechanism might contribute to the inactivation of the Hippo pathway in both settings upon *Ptp10D* knockdown. It is also possible that Ptp10D might be regulating the Hippo pathway directly; for example, because Ptp10D is a phosphatase, it might function to dephosphorylate components of the Hippo pathway, such as Expanded (a Hippo pathway upstream activator). Expanded is phosphorylated by Casien Kinase 1 and then targeted for degradation by the Slimb-β-TrcP ubiquitin ligase (Fulford et al., 2019). If Ptp10D can dephosphorylate Expanded, its

knockdown would be expected to lead to higher levels of Expanded phosphorylation and its subsequent degradation, therefore resulting in greater suppression of the Hippo pathway. Alternatively, Ptp10D might regulate signaling pathways that regulate Hippo signaling, such as the Hedgehog (Kagey et al., 2012) or Decapentaplegic pathway (Oh and Irvine, 2011), by dephosphorylation of key components in these pathways.

## *Ptp10D* knockdown rescues the adult eye phenotype of moderate *RhoGEF2*$^{OE}$ mosaics

Interestingly, although moderate *RhoGEF2*$^{OE}$ *Ptp10D*$^{KD}$ clones were increased in size within the eye-antennal imaginal discs, the adult eye phenotype was largely rescued and presented with a less-severe phenotype than moderate *RhoGEF2*$^{OE}$ (plus *Dicer2*) mosaics. This is different to what has been observed so far with loss of the polarity regulators Scrib and Dlg, where *Ptp10D* knockdown increased the severity of the adult eye phenotype (Yamamoto et al., 2017). Furthermore, although no necrotic tissue was found in moderate *RhoGEF2*$^{OE}$ or *RhoGEF2*$^{OE}$ *Ptp10D*$^{KD}$ mosaic adult eyes, some of the *RhoGEF2*$^{OE}$ adult eye mosaics had protrusions present that were not observed upon *Ptp10D* knockdown. In some *RhoGEF2*$^{OE}$ mosaic adult eyes, it seemed like parts of the antenna were present in the eye structure, suggesting that transdetermination events might be occurring, as has been previously observed upon elevation of Rho and/or Rac signaling (Brumby et al., 2011). Transdetermination has been shown to involve ectopic Wingless signaling (McClure and Schubiger, 2007; Schubiger et al., 2010), so it is possible that *Ptp10D* knockdown might inhibit this pathway, thereby rescuing the adult eye defects. A more detailed understanding of the signaling pathways regulated by Ptp10D and RhoGEF2, should provide insight into the rescue the *RhoGEF2*$^{OE}$ mosaic adult eye defect seen upon *Ptp10D* knockdown.

## Involvement of the apical protein Crb in elimination of *RhoGEF2*$^{OE}$ cytoskeletal deregulated clones

As Crb is a known regulator of the Hippo pathway (reviewed in Richardson and Portela, 2017) and, like Ptp10D, is localized to the apical membrane, we investigated the involvement of *crb* in *RhoGEF2*$^{OE}$ clone elimination. We showed that *crb* loss in *RhoGEF2*$^{OE}$ clones partially rescued their small size, which indicates that Crb is playing a role in the elimination of *RhoGEF2*$^{OE}$ clones. Furthermore, a significant decrease in cell death was observed in *RhoGEF2*$^{OE}$ *crb*$^{-/-}$ clones when compared to *RhoGEF2*$^{OE}$ clones alone. These findings show that Crb plays an important role in the elimination of high overexpression *RhoGEF2* clones.

Surprisingly, loss of *crb* reduced JNK signaling in high *RhoGEF2*$^{OE}$ clones, suggesting that JNK upregulation is Crb dependent and, therefore, JNK downregulation might contribute to the increased survival of *crb*$^{-/-}$ *RhoGEF2*$^{OE}$ clones. Activation of Rho1, Rok and Myosin II contribute to JNK pathway activation in *RhoGEF2*$^{OE}$ eye-antennal disc clones (Khoo et al., 2013), and Crb might function through this same axis to regulate JNK activity. Furthermore, loss of *crb* in high *RhoGEF2*$^{OE}$ clones rescued the low levels of pERK and Diap1 in *RhoGEF2*$^{OE}$ clones. These results suggest that the mechanism by which *RhoGEF2*$^{OE}$ *crb*$^{-/-}$ clones increase in size involves elevated EGFR-Ras signaling and reduced Hippo signaling but, given that JNK signaling was normalized and F-actin accumulation reduced, that it is independent of elevated JNK signaling and F-actin accumulation. The reduced F-actin accumulation in *RhoGEF2*$^{OE}$ *crb*$^{-/-}$ clones might be a result of low EGFR-Ras and JNK signaling, as in *scrib*$^{-/-}$ clones these pathways act synergistically

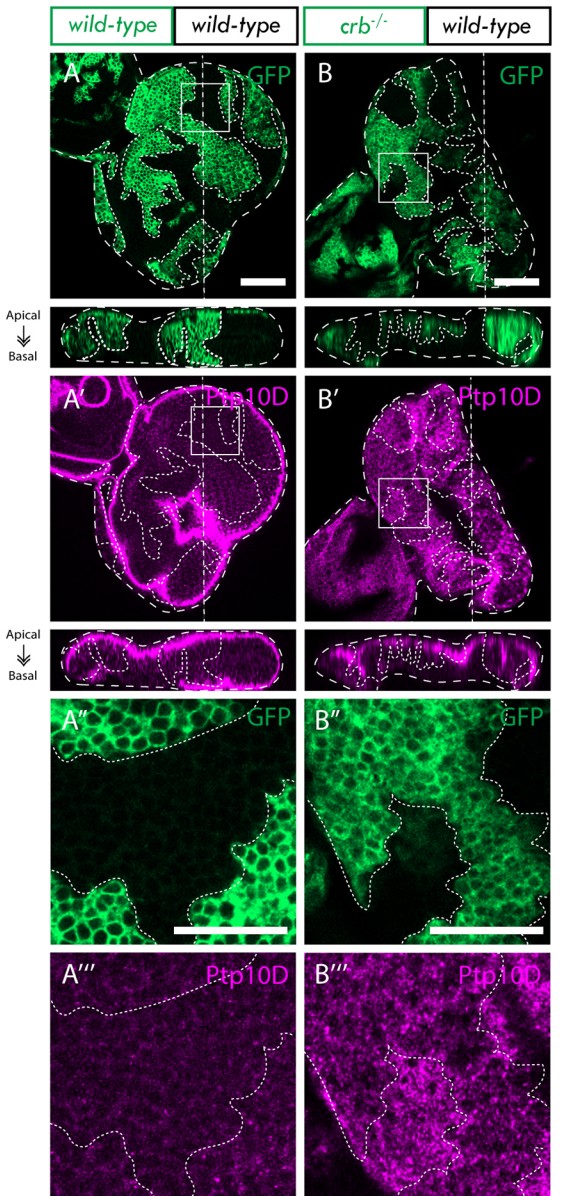

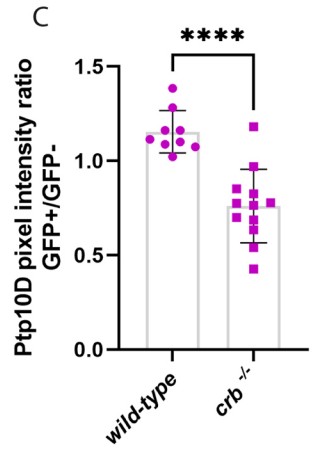

**Fig. 7. Ptp10D is downregulated in *crb* mutant clones.** (A,B) Eye discs of *ey-FLP-MARCM*-induced mosaics (clones are marked by the presence of GFP and wild-type tissue is unmarked): (A) wild-type; (B) *crb*$^{-/-}$; (A′,B′) Ptp10D immuno stains from A,B; white squares indicate magnified sections shown in panels below; (A″,B″) Magnified sections from A,B; (A‴,B‴) Ptp10D immunostains from A″,B″. (C) Quantification of Ptp10D immunostain pixel intensity ratio between GFP+ and GFP− clones [mean±s.d.; wild-type (*n*=9); *crb*$^{-/-}$ (*n*=12)]. ****$P<0.0001$ (two-tailed unpaired *t*-test, $P<0.05$). Images below show an *xz* cross section of the corresponding eye-antennal disc from the apical (top) to basal (bottom) edge, with the position of the chosen *xz* sections indicated by a vertical dotted line in the *xy* images. Note that folding at the edges of the discs can result in Ptp10D being observed basally in the *xz* sections in these regions, despite the staining still being localized at the apical membrane. Dotted lines surrounding discs or clones illustrate disc/clone boundaries. Scale bars: 50 μm (A,A′,B,B′); 20 μm (A″,A‴,B″ and B‴).

in promoting F-actin accumulation (Yamamoto et al., 2017). The reduced F-actin accumulation in high *RhoGEF2*$^{OE}$ *crb*$^{-/-}$ clones compared to *RhoGEF2*$^{OE}$ clones alone is consistent with these clones being less cyst-like in structure, because these clone structures form upon accumulation of actin cytoskeletal proteins (Bielmeier et al., 2016). Consistent with this, loss of *crb* in *RhoGEF2*$^{OE}$ clones, as in *RhoGEF2*$^{OE}$ *Ptp10D*$^{KD}$ clones, rescued the severe *RhoGEF2*$^{OE}$ eye phenotype, which is likely due to deregulation in actomyosin dynamics.

How is Crb functioning to contribute to the elimination of high *RhoGEF2*$^{OE}$ clones? It has been previously described that the FERM-binding motif (FBD) of Crb controls actinmyosin dynamics (Flores-Benitez and Knust, 2015), and actinmyosin regulators have been shown to control Hippo signaling (Deng et al., 2015; Fernández et al., 2011; Fletcher et al., 2015; Gaspar et al., 2015; Külshammer and Uhlirova, 2013; Wong et al., 2015). Therefore, upon *crb* loss in high *RhoGEF2*$^{OE}$ clones, a change in regulation of actinmyosin dynamics mediated by Crb might explain the decreased F-actin accumulation observed, as well as contribute to the

impairment of the Hippo pathway. Additionally, the FBD of Crb also binds to the Hippo pathway regulator, Expanded, regulating its stability and the activity of the Hippo pathway (Chen et al., 2010; Grzeschik et al., 2010; Robinson et al., 2010), and therefore *crb* loss would be expected to directly result in impaired Hippo signaling in the high *RhoGEF2*$^{OE}$ clones, accounting for their increased survival. Consistent with our observations of the importance of Crb in *RhoGEF2*$^{OE}$ cell competition, we observed an upregulation of Crb in *RhoGEF2*$^{OE}$ clones (Fig. S9), which might lead to increased Hippo pathway activation.

### Are Crb and Ptp10D acting on the same pathway?
We found that *crb*$^{-/-}$ reduced the abundance and apical membrane localization of Ptp10D and therefore hypothesized that the effect observed upon loss of *crb* in *RhoGEF2*$^{OE}$ clones might be a direct consequence of deregulated Sas-Ptp10D signaling. The major commonality between *crb* loss and *Ptp10D*$^{KD}$ in *RhoGEF2*$^{OE}$ clones was the effect on Diap1. Diap1 downregulation in high *RhoGEF2*$^{OE}$ clones was rescued to wild-type levels by loss of *crb*,

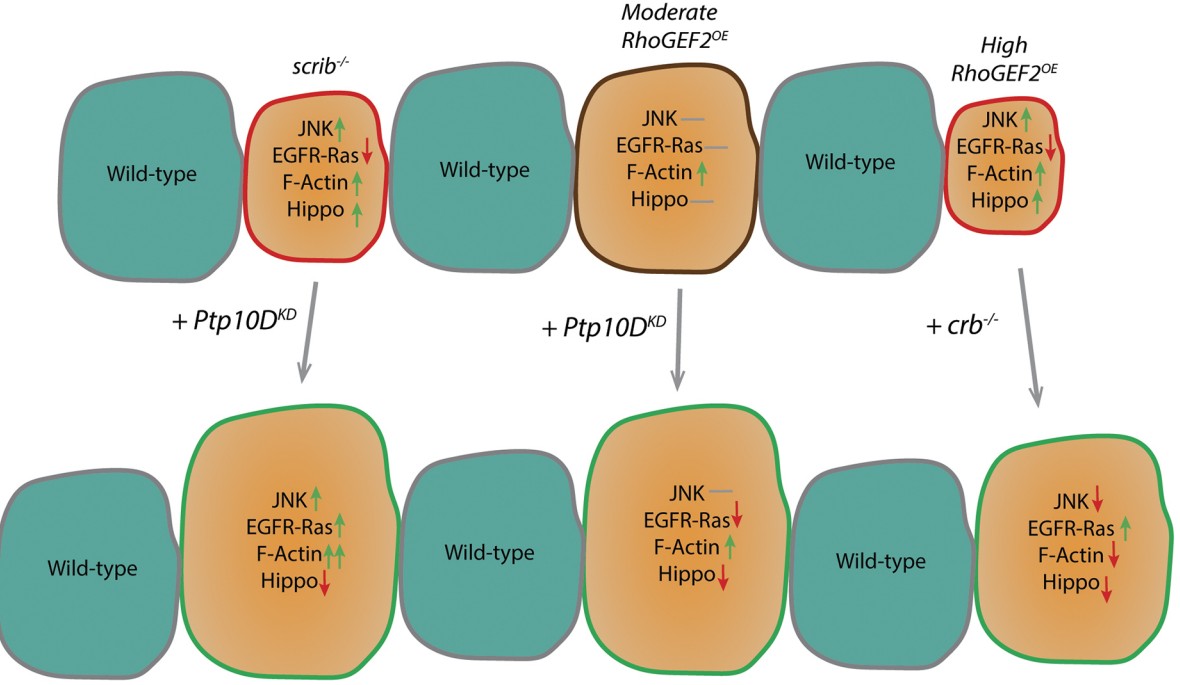

**Fig. 8. Summary of cell signaling pathway activity upon the different genetic alterations studied in this manuscript.** Green arrows indicate upregulation, red arrows indicate downregulation, grey horizontal line indicate no change observed or not investigated. A red clone outline signifies that this clone becomes a loser clone when surrounded by wild-type cells. A green clone outline indicates that these clones survive and grow bigger, a brown line indicates that no clone size change is observed. Clone sizes are shown to represent relative sizes for each condition. $scrib^{-/-}$ clones are eliminated by cell competition through upregulation of JNK, which is acting through its pro-apoptotic role since EGFR-Ras signaling is blocked by Ptp10D. F-actin is upregulated and so is Hippo pathway signaling. Upon Ptp10D knockdown, EGFR-Ras signaling is upregulated and acts synergistically with JNK resulting in F-actin hyperaccumulation and a reduction of Hippo pathway signaling (Yamamoto et al., 2017). Moderate RhoGEF2 overexpression clones show elevated active Caspase-3 but are not reduced in size, and show elevated F-actin accumulation, but JNK, EGFR-Ras and Hippo signaling are not affected. Upon Ptp10D knockdown, the Hippo pathway is downregulated, and the EGFR-Ras is slightly downregulated. Given that Hippo signaling is decreased without elevated EGFR-Ras signaling upon Ptp10D knockdown, this indicates that Ptp10D knockdown may be inhibiting the Hippo pathway more directly in moderate $RhoGEF2$ overexpressing clones to result in increased clone survival. High RhoGEF2 overexpression clones show F-actin accumulation and reduced EGFR-Ras signaling and are eliminated through a JNK dependent mechanism and Hippo upregulation. Upon $crb$ loss, JNK signaling, Hippo signaling and F-actin accumulation are reduced, and EGFR-Ras signaling is upregulated.

whereas $Ptp10D^{KD}$ led to increased Diap1 levels in moderate $RhoGEF2^{OE}$ clones compared to what was found in wild-type controls. Given that loss of $crb$ leads to a downregulation of the level and localization of Ptp10D to the apical membrane, it is possible that the effect of $crb$ loss on the Hippo signaling pathway is dependent, to some extent, on reduced Ptp10D activity. As mentioned above, Ptp10D might also contribute to the regulation of Expanded activity through its phosphatase activity, preventing the phosphorylation-mediated degradation of Expanded and promoting Hippo activity. Owing to technical issues, we were not able to knockdown $Ptp10D$ in high $RhoGEF2^{OE}$ clones and hence could not assess whether $Ptp10D$ knockdown phenocopies other effects of $crb$ loss in high $RhoGEF2^{OE}$ clones. Further investigation is needed to determine whether $Ptp10D$ knockdown has the same effect in high $RhoGEF2^{OE}$ clones as it does in $scrib$ mutant clones (Yamamoto et al., 2017) or acts more like $crb$ loss in this context.

The Sas-Ptp10D system had previously only been described to have a role in cell competition for the elimination of polarity-deficient cells. Interestingly, here we show that Sas and Ptp10D relocate to the lateral membrane in $RhoGEF2^{OE}$ clones, as occurs with $scrib^{-/-}$ clones (Yamamoto et al., 2017), and that $Ptp10D^{KD}$ in $RhoGEF2^{OE}$ clones increases $RhoGEF2^{OE}$ clonal growth. These two findings indicate that Sas-Ptp10D signaling plays a role in restricting the clonal growth of $RhoGEF2^{OE}$ cells, where the actin cytoskeleton is deregulated but cell polarity is not grossly affected,

and, therefore, that polarity loss is not a prerequisite for Sas-Ptp10D activation. It is possible that in other forms of cell competition where a precise mechanism is unknown, such as in $Stat92E$-overexpressing clones (Rodrigues et al., 2012; Zoranovic et al., 2013), Sas-Ptp10D signaling might also play a role, and it will be important for future endeavors to investigate this question. Additionally, it will be important to determine whether a role for Crb exists in polarity-impaired cell competition, as well as other cell competition scenarios.

## MATERIALS AND METHODS

### *Drosophila* stocks

The following fly stocks were used: *FRT82B* (BDSC) #2035, $scrib^{1}$, $crb^{11A22}$ (Tepass and Knust, 1990), *UAS-myr-RFP* (BDSC) #7118, *UAS-Ptp10D-RNAi* (VDRC) #1101, *UAS-Ptp10D-RNAi* (VDRC) #1102, *UAS-sas-RNAi* (VDRC) #39086, *UAS-RhoGEF2* (Padash Barmchi et al., 2005) 3R MARCM stock with eyeless promoter [*ey-FLP, UAS-mCD8-GFP;; tub-GAL4 FRT82B tub-GAL80/TM6B* (MARCM 82B)] (Lee and Luo, 2001), and 3R Reverse MARCM stock with eyeless promoter [*y-, w-, eyFLP2; Act-GAL4, UAS-GFP; tub-GAL80, FRT82B, scrib1/TM6B*] (Xianjue Ma and J. E. La Marca). See Table S1 for genotypes of flies used for each figure.

### Husbandry conditions and food recipes
Stocks were maintained at room temperature (RT). Crosses were undertaken at 25°C in an incubator with a 12-h-light–12-h-dark cycle (unless otherwise indicated). Flies were fed with standard semolina-molasses-agar medium

supplemented with live yeast or a low protein diet when indicated (see Table S2).

## Immunohistochemistry – sample preparation and antibodies

Wandering third-instar larvae (L3) were picked for all experiments, and for each experiment all larvae were of equivalent size. L3 eye-antennal discs were dissected in 1× phosphate-buffered saline (PBS) (Amresco #703) and fixed in 4% paraformaldehyde (Alfa Aesar #43368) in PBS with 0.3% Triton X-100 (Amresco #0694) (PBST) for 20-30 min at RT (note: when using the pERK antibody, tissues were fixed in 8% PFA). After fixation, tissues were washed and permeabilized three times for at least 10 min in PBS with 0.3% Triton X-100 (PBST), and then blocked for at least 60 min at RT in PBST containing 5% bovine serum albumin (BSA) (Sigma-Aldrich #A2153). Unless otherwise indicated, tissues were then incubated overnight at 4°C, in dilutions of primary antibodies in PBST containing 5% BSA. The next day, tissues were washed three times for at least 10 min in PBST, then incubated in secondary antibody in PBST containing 5% BSA at RT for 60–120 min. Before mounting, tissues were washed in PBST again three times for minimum 10 min. All samples were mounted in VECTASHIELD® Antifade Mounting Medium with DAPI (#H-1200-10). For cytoskeleton structure visualization (F-actin), Rhodamine–phalloidin fluorescently tagged with RFP was used 1:250 (Thermo Fisher Scientific #R415). Utilized primary antibodies were: mouse anti-Ptp10D (1:500, DSHB, 8B22F5), rabbit anti-Sas (1:200, Elisabeth Knust, Max Planck Institute, Dresden, Germany), rabbit anti-cleaved Caspase-3 (1:100, Cell Signaling Technologies #9661), mouse anti-MMP1 (1:100, DSHB, #3B8D12, #5H7B11 and #3A6B4), mouse anti-Diap1 (1:100, Bruce Hay, California Institute of Technology, Pasadena, CA, USA), rabbit anti pERK (1:200, Cell Signaling technologies, #4370), and rat anti-Crumbs (1:400, Elisabeth Knust). Utilized secondary antibodies were: anti-mouse IgG, Alexa Fluor 568 (1:500, Thermo Fisher Scientific, #A11004), anti-rabbit IgG, Alexa Fluor 633 (1:500, Thermo Fisher Scientific, #A21070), and anti-rat IgG, Alexa Fluor 568 (1:500, Thermo Fisher Scientific #A11077).

## Immunohistochemistry – imaging

Samples were imaged using either a Zeiss LSM 780 or Zeiss LSM 800 laser scanning confocal microscope, and images were processed using Zen 3.0 SR software (Zeiss). Images were generally taken as Z-stacks, but all images shown in this study represent single planes of the Z-stacks, unless otherwise specified. Laser intensity and gain was unchanged within each experimental group.

## Image analysis – clone volumes

Clone volume quantification was analyzed specifically in the eye primordium (the antennal primordium was not included). Quantification of tissue volume was analyzed with Imaris software (Bitplane). This software allows for the creation of a 3D structure of the eye imaginal disc by combining all the Z-stack images. Using this structure, the volume of the GFP+ tissue could be determined by generating a mask of the GFP+ tissue. Similarly, the total volume of the eye disc was determined by generating a mask of the DAPI+ tissue. Where thresholding was necessary to differentiate between the tissue positive or negative for these different markers, automatic threshold values chosen by Imaris were used so as not to bias the data. Then, to determine the GFP+ tissue to total tissue ratio the following equation was used [GFP+ volume (µm$^3$)/DAPI+ volume (µm$^3$)]×100=% GFP+ tissue.

## Image analysis – pixel intensity ratios

To investigate differences in expression levels or activity of different genes/pathways, pixel intensity ratios were determined using FIJI software (Schindelin et al., 2012) were performed. The 'mean' pixel intensity of a specific immunostain within the GFP+ clone tissue was measured using the square selection tool and divided by the mean intensity measured in an identical square selection in an adjacent GFP– region. This was repeated 3–5 times per disc, using different clones each time. The number of analyzed discs is specified in the relevant figure legends. Clones to be assessed were selected based on their positioning within the tissue, with those near the tissue borders generally avoided due to unusually high accumulations of stains, such as for F-actin and Crb. F-actin quantifications were performed in apical Z-stacks.

## Image analysis – spot density ratios

To measure puncta of cleaved Caspase-3, the 'Spots' tool in Imaris software was utilized to automatically count the total number of positively stained cells. Thresholding was used to only include spots of 3 µm (or larger) in diameter. The 'Filter' tool was utilized to automatically quantify the number of positively stained cells specifically localized within GFP+ tissue. To normalize the number of positively stained cells to the amount of tissue, the following calculation was used: number of stained cells in GFP+ tissue/total volume of GFP+ tissue=A, number of stained cells in GFP– tissue/total volume of GFP– tissue=B, and then, A/B=spot density ratio within GFP+ tissue.

## Adult eye imaging and analysis

Unconscious flies were placed in 1.5 ml microcentrifuge tubes, and euthanized at −20°C for at least 30 min. Flies were then positioned for imaging using a Petri dish with a raised section to allow for consistent positioning of the head. Eyes were then imaged using a 'Fully Automated Fluorescence Stereo Microscope Leica M205 FA' and Leica Application Suite v4.2.0 software (Leica). All flies were imaged the same day they were euthanized to avoid tissue dehydration.

## Eye size quantifications

To analyse for differences in the eye sizes, FIJI software was utilized to measure the area of the eye and the total area of the fly head in pixels. Then, the ratio of the areas using the following equation was determined: area ratio=eye area/head area.

## Adult eye severity phenotyping

To analyze potential increases in the severity of the adult eye phenotype, a 'phenotype severity' score was given, where 1 indicates a wild-type-like phenotype, 2 indicates a rough eye with some tissue rugosities, 3 indicates a rougher eye with potentially some necrotic speckles, and 4 indicates a strongly rough eye with necrotic tissue or epithelial protrusions (see Fig. S10). Scoring was performed by an investigator who was aware of the experimental conditions.

## Statistical analyses

Statistical analyses were performed and graphs generated, using Prism 9 (GraphPad). We performed a D'Agostino and Pearson normality test. Data that followed a normal distribution were analyzed using a two-tailed unpaired t-test. For tests requiring multiple comparisons, a one-way ANOVA followed by Tukey's multiple comparison test was used. Data that did not pass the normality test were analyzed using a two-tailed Mann–Whitney U-test or, where multiple comparisons were required, a Kruskal–Wallis test with Dunn's post-hoc test. P<0.05 was considered statistically significant.

### Acknowledgements

We thank Peter Burke for help maintaining the Drosophila stocks, and all other members of our lab for constructive discussions. We thank the LIMS Bioimaging Facility for microscopy equipment and technical support. We thank the Australian Drosophila research community, the BDSC and the VDRC for providing fly stocks, OzDros for stock importation services, and FlyBase for the wealth of information. For antibodies, we thank various laboratories (as indicated in the Materials and Methods) and the Developmental Studies Hybridoma Bank, which was created by the National Institute of Child Health and Human Development of the National Institutes of Health and is based at The University of Iowa, for supplying antibodies.

### Competing interests

The authors declare no competing or financial interests.

### Author contributions

Conceptualization: H.E.R.; Data curation: N.F.-L.; Formal analysis: N.F.-L.; Funding acquisition: H.E.R.; Investigation: N.F.-L.; Methodology: N.F.-L., M.P.; Project administration: H.E.R.; Supervision: M.P., J.E.L.M., H.E.R.; Writing – original draft: N.F.-L., H.E.R.; Writing – review & editing: M.P., J.E.L.M., H.E.R.

### Funding

N.F.-L. was supported by a scholarship from La Trobe University. J.E.L.M. was supported by an Australian Research Council Discovery grant to H.E.R.

(DP170102549). M.P. was supported by a National Health and Medical Research Council grant to H.E.R. (APP1160025). H.E.R. was supported by the La Trobe Institute for Molecular Science and La Trobe University. Open Access funding provided by La Trobe University. Deposited in PMC for immediate release.

**Data and resource availability**
All relevant data can be found within the article and its supplementary information.

**Peer review history**
The peer review history is available online at https://journals.biologists.com/jcs/lookup/doi/10.1242/jcs.264377.reviewer-comments.pdf

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
