## [Peer Review File · Journal of Cell Science]

RhoGEF2 overexpression induces cell competition dependent on Ptp10D, Crumbs and the Hippo signaling pathway.

Natasha Fahey-Lozano, Marta Portela, John E. La Marca and Helena E. Richardson
DOI: 10.1242/jcs.264377

Editor: Michael Way

Review timeline

Submission to Review Commons:	8 May 2025
Submission to Journal of Cell Science:	12 August 2025
Accepted:	13 August 2025

Reviewer 1

Evidence, reproducibility and clarity

Summary: findings and key conclusions

Epithelial cell competition in larval imaginal discs involves signaling with the Sas ligand and Ptd10D receptor. In wild type cells both are typically found at the apical surface, but relocalize to the lateral cortex at the winner-loser interface. Ptd10D activation leads to reduced Ras signaling, increased pro-apoptotic Jnk signaling and consequently the elimination of loser cells. In the manuscript the authors address the role of the actin cytoskeleton in the context of the signaling controlling cell elimination in *Drosophila* larval eye imaginal discs. They interfere by clonal overexpression of the guanyl nucleotide exchange factor RhoGEF2 (RG2), which has previously been shown to induce dominant gain-of-function phenotypes by activation of Rho signaling. In this context the requirement of and genetic interactions with the other pathways implicated in cell elimination is tested. They find that RG2 induced cell elimination depends on Ptd10D, Hippo signaling and Crumbs.

Major comments: claims and conclusions

The experimental setting, using clonal analysis in imaginal discs, is straight-forward and well-established, including quantification of clone size and comparison of phenotypes. The presented data are of high quality and thus the direct conclusions are fully supported by the data as long as they refer to the actual experimental interference. What is not supported by the data is the generalization of the conclusions, i. e. that RG2 overexpression would be equivalent to Actin cytoskeletal deregulation. This equivalence is expressed in the title "Actin cytoskeletal deregulation, caused by RhoGEF2 overexpression.." and the summary "that actin cytoskeleton deregulated cells (as induced by RhoGEF2 overexpression (RhoGEF2OE))..". In my view such an equivalence is not justified. There is no doubt that RG2 overactivation affects the actin cytoskeleton in multiple ways, such as contractility via MyoII or polymerization via Dia, among others. There is also no doubt that other pathways are also directly or indirectly affected beside the actin cytoskeleton. The authors do not present data showing the specificity of RG2 overexpression. For example, the authors could investigate the phenotype and genetic interaction with an alternative way of interference, independent of RG2, of the actin cytoskeleton to support their conclusion.

There is a second assumption, which may not be justified, that the function of the cytoskeleton would be generally downstream of cell polarity, see abstract l24 "triggering cytoskeletal deregulation (which occurs downstream of cell polarity disruptions)..". There are certainly cytoskeletal activities such as cell shape changes that mediate the execution of cell elimination.

However interfering with the cortical cytoskeleton also affect the distribution of cortical polarity proteins. The authors do not present data to demonstrate the specificity of RG2 overexpression concerning a function downstream of cell polarity.

Significance

The study establishes genetic interactions and dependencies concerning cell elimination following a very specific experimental interference of RG2 overexpression. It remains unclear, however, to which degree these genetic interactions contribute to controlling cell competition in situations that are physiologically relevant. The generalization of RG2 overexpression as a specific test the function of the actin cytoskeleton is an interpretation not supported by the presented data and the experimental set up.

Reviewer 2

Evidence, reproducibility and clarity

Summary:

In the manuscript "Actin cytoskeletal deregulation, caused by RhoGEF2 overexpression, induces cell competition dependent on Ptp10D, Crumbs, and the Hippo signaling pathway", Natasha et al. investigate how actin cytoskeletal deregulation drives cell competition in the *Drosophila* eye disc. By overexpressing RhoGEF2 to induce cytoskeletal disruption and utilizing genetic knockdowns of various candidate genes, the authors examine the spatial distribution and interaction between normal and deregulated cell populations. Their findings demonstrate that cell competition and clone elimination in this context are dependent on *sas*, Ptp10D, *scrib*, and components of the Hippo signaling pathway. The study is well executed and provides a potentially impactful contribution to the field. The experimental design is solid, and the conclusions are generally well supported by the data. Only minor revisions are needed to strengthen clarity and presentation. Specific suggestions and comments regarding significance are listed below.

Major comments:

There is a discrepancy between the representative images (Fig. 3A-C') and the quantification in Fig. 3J. The statistical analysis may be limited by small sample size or suboptimal test choice (e.g., Kruskal-Wallis vs. ANOVA). Increasing the sample size and reassessing the statistical approach could strengthen this otherwise well-executed section.

A minor concern relates to the interpretation and consistency of the statistical analyses used. For example, in Figure 5I, both the Kruskal-Wallis test and an unpaired t-test were used, with the authors stating that the t-test was applied specifically to compare wild-type and *crb*^{-/-} clones ($p = 0.0147$). However, in the adjacent panel (Figure 5J), only a one-way ANOVA was used. This inconsistency may give the impression that the choice of statistical test in Figure 5I was influenced by a lack of significance with the Kruskal-Wallis test, rather than by experimental design. Unifying the statistical approach within related panels would improve clarity and minimize potential reader misinterpretation. Additionally, some of the statistical tests applied may not fully align with the underlying data distributions. Statistical methods used in parts of the manuscript may need to be reevaluated, and the rationale for their selection should be clarified in the text.

In the section on how *crb*^{-/-} affects actin distribution and accumulation within the tissue (Figure 6H' and Supplementary Figure 5), it appears that F-actin may accumulate more prominently in cytoplasmic regions rather than at cell-cell junctions under *crb*^{-/-} conditions. However, due to the current level of magnification, it is difficult to determine the precise subcellular localization. Although this question is somewhat tangential to the main focus of the manuscript and not essential for publication, it could be valuable, if the authors included a few higher-magnification images showing F-actin distribution in RhoGEF2OE *Dicer2*, RhoGEF2OE Ptp10D KD, and RhoGEF2OE *crb*^{-/-} conditions. Including these in the supplementary figures could help clarify how actin cytoskeletal regulation is affected.

In Figure 6H', the Diap1 signal in the RhoGEF2OE condition appears non-uniform, with noticeably weaker intensity on the left side of the image and stronger signal on the right. This asymmetry is

not observed in the RhoGEF2OE *crb*^{-/-} condition shown in Figure 6K'. It is unclear whether this pattern reflects a biological phenomenon consistently observed in RhoGEF2OE tissues or if it might result from technical factors such as uneven mounting or imaging. To prevent potential misinterpretation, we recommend clarifying this point, providing additional representative images if available, or replacing the current image with one that more clearly reflects the typical expression pattern.

In Fig. 3B', cleaved Caspase-3 appears localized to specific regions at the WT/RhoGEF2OE interface, suggesting spatial bias in Ptp10D-dependent elimination. This raises important questions about what determines regional susceptibility—are certain tissue conditions or cell states more prone to apoptosis in this context?

Figure 3 raises the question of whether RhoGEF2OE-induced, actin-deregulated clones undergo dynamic changes, such as expanding or regressing, over the course of the larval stage. Such temporal variability could influence GFP⁺ clone size and the expression of apoptotic markers like cleaved Caspase-3 and Diap1. The stated use of the L3 stage, which spans ~48 hours (Tennessee & Thummel, 2011), lacks sufficient temporal resolution.

Clarifying the timing of dissection and fixation relative to clone induction would improve interpretation of clone behavior and marker dynamics.

Minor comments:

GFP signal appears weaker in the wild-type group compared to experimental conditions, raising the question of whether image processing (e.g., contrast and color balance) was applied uniformly and if this difference reflects true variation in expression.

For Figures 2, 3, and 5, including representative images for each eye phenotype category would clarify the scoring criteria. In Figure 5, the use of a "2.5" category in the main figure should be explained—does it correspond to category 3 or indicate an intermediate phenotype? In Figure 5I, the y-axis range (0-150%) is broader than needed; adjusting it to 0-100% would better reflect the data and improve clarity.

The sentence from line 343- 348 is long and challenging to follow. Missing the Figure number on Line 286.

Significance

This study is well executed and rigorously addresses previously reported variations in phenotypic outcomes across laboratories. Beyond clarifying the role of Ptp10D in cell competition, the authors establish RhoGEF2 overexpression as a reliable method to induce cell competition and identify key molecular players involved in this process. This work represents a meaningful advance by introducing novel approaches and deepening understanding of known factors in clone elimination. The mosaic RhoGEF2 overexpression technique developed in this study provides a valuable tool for investigating cell-cell interactions at the tissue level, with broad applicability in basic research. This approach holds particular promise for probing

Author response to reviewers' comments

Reviewer #1 (Evidence, reproducibility and clarity (Required)):

Summary: findings and key conclusions

Epithelial cell competition in larval imaginal discs involves signaling with the Sas ligand and Ptd10D receptor. In wild type cells both are typically found at the apical surface, but relocalize to the lateral cortex at the winner-loser interface. Ptd10D activation leads to reduced Ras signaling, increased pro-apoptotic Jnk signaling and consequently the elimination of loser cells. In the manuscript the authors address the role of the actin cytoskeleton in the context of the signaling controlling cell elimination in *Drosophila* larval eye imaginal discs. They interfere by clonal overexpression of the guanyl nucleotide exchange factor RhoGEF2 (RG2), which has previously been shown to induce dominant gain-of-function phenotypes by activation of Rho signaling. In this context the requirement of and genetic interactions with the other pathways implicated in cell

elimination is tested. They find that RG2 induced cell elimination depends on PtD10D, Hippo signaling and Crumbs.

Major comments: claims and conclusions

The experimental setting, using clonal analysis in imaginal discs, is straight-forward and well-established, including quantification of clone size and comparison of phenotypes. The presented data are of high quality and thus the direct conclusions are fully supported by the data as long as they refer to the actual experimental interference. What is not supported by the data is the generalization of the conclusions, i. e. that RG2 overexpression would be equivalent to Actin cytoskeletal deregulation. This equivalence is expressed in the title "Actin cytoskeletal deregulation, caused by RhoGEF2 overexpression.." and the summary "that actin cytoskeleton deregulated cells (as induced by RhoGEF2 overexpression (RhoGEF2OE))...". In my view such an equivalence is not justified. There is no doubt that RG2 overactivation affects the actin cytoskeleton in multiple ways, such as contractility via MyoII or polymerization via Dia, among others. There is also no doubt that other pathways are also directly or indirectly affected beside the actin cytoskeleton. The authors do not present data showing the specificity of RG2 overexpression. For example, the authors could investigate the phenotype and genetic interaction with an alternative way of interference, independent of RG2, of the actin cytoskeleton to support their conclusion.

There is a second assumption, which may not be justified, that the function of the cytoskeleton would be generally downstream of cell polarity, see abstract l24 "triggering cytoskeletal deregulation (which occurs downstream of cell polarity disruptions)..". There are certainly cytoskeletal activities such as cell shape changes that mediate the execution of cell elimination. However interfering with the cortical cytoskeleton also affect the distribution of cortical polarity proteins. The authors do not present data to demonstrate the specificity of RG2 overexpression concerning a function downstream of cell polarity.

Response: We apologise for our phrasing of the title and the sentence in the summary that suggests that it is the actin cytoskeleton disruption caused by *RhoGEF2* overexpression that is responsible for the effects on cell competition. We have rephrased the title and edited the text to avoid such an inference.

With regard to the reviewer's second concern regarding the link between cell polarity disruption and actin cytoskeletal deregulation, there is indeed evidence that this occurs.

There are numerous examples of how cell polarity regulators affect the actin cytoskeleton in both *Drosophila* and mammalian cells (reviewed by Humbert et al., 2015, DOI 10.1007/978-3-319-14463-4_4). Indeed, in our previous paper (Brumby et al., 2011. PMID: 21368274), we found genetic evidence that the knockdown of the polarity regulator, *dlg*, cooperates with activated Ras (RasACT) to produce a hyperplastic eye phenotype, and that this phenotype is rescued by knockdown of actin cytoskeletal regulators like RhoGEF2 or Rho. This data suggests that these actin cytoskeleton regulators act downstream of cell polarity disruption to cooperate with RasACT. Furthermore, another study has shown that the activation of Myosin II is increased in *scrib* mutants and impairs Hippo pathway signaling, and is also required for the cooperation of *scrib* mutants with RasACT (Külshammer, et al., 2013. PMID: 23239028). Consistent with this finding, we have previously shown that RhoGEF2 acts via Rho, Rok, and Myosin II activation in cooperation with RasACT (Khoo et al., 2013. PMID: 23324326). Furthermore, another cell polarity regulator, Lgl, binds to and negatively regulates Myosin II function in *Drosophila* (Strand et al., 1994. PMID: 7962095; Betschinger et al., 2005. PMID: 15694314). Moreover, *Drosophila* Scrib and Dlg bind to GUK-holder/NHS1 (Nance-Horan syndrome-like 1), which is a regulator of the WAVE/SCAR-ARP2/3-branched F-actin pathway, and this interaction is required for epithelial tissue development (Caria et al., 2018. PMID: 29378849). Thus, although cell polarity gene loss can affect the actin cytoskeleton by different means, and RhoGEF2 can activate Rho to regulate various actin cytoskeletal effectors (Limk, Dia, PKN, Rok), what they have in common is the activation of Myosin II. To make this clearer, we have now added brief sections to the introduction and Discussion highlighting and contextualising evidence for the effect of cell polarity disruption on the actin cytoskeleton.

Reviewer #1 (Significance (Required)):

The study establishes genetic interactions and dependencies concerning cell elimination following a very specific experimental interference of RG2 overexpression. It remains unclear, however, to

which degree these genetic interactions contribute to controlling cell competition in situations that are physiologically relevant. The generalization of RG2 overexpression as a specific test the function of the actin cytoskeleton is an interpretation not supported by the presented data and the experimental set up.

Response: Although *RhoGEF2* overexpression does lead to actin cytoskeletal disruption via Rho effectors, the reviewer is correct that we do not know whether it is the actin cytoskeleton disruption per se that is involved in triggering cell competition. We have edited the text accordingly.

Reviewer #2 (Evidence, reproducibility and clarity (Required)):

Summary:

In the manuscript "Actin cytoskeletal deregulation, caused by RhoGEF2 overexpression, induces cell competition dependent on Ptp10D, Crumbs, and the Hippo signaling pathway", Natasha et al. investigate how actin cytoskeletal deregulation drives cell competition in the *Drosophila* eye disc. By overexpressing RhoGEF2 to induce cytoskeletal disruption and utilizing genetic knockdowns of various candidate genes, the authors examine the spatial distribution and interaction between normal and deregulated cell populations. Their findings demonstrate that cell competition and clone elimination in this context are dependent on *sas-Ptp10D*, *scrib*, and components of the Hippo signaling pathway. The study is well executed and provides a potentially impactful contribution to the field. The experimental design is solid, and the conclusions are generally well supported by the data. Only minor revisions are needed to strengthen clarity and presentation. Specific suggestions and comments regarding significance are listed below.

Major comments:

There is a discrepancy between the representative images (Fig. 3A-C') and the quantification in Fig. 3J. The statistical analysis may be limited by small sample size or suboptimal test choice (e.g., Kruskal-Wallis vs. ANOVA). Increasing the sample size and reassessing the statistical approach could strengthen this otherwise well-executed section.

Response: The data is normally distributed, so we have repeated the analysis using a one-way ANOVA (instead of the Kruskal-Wallis test - Initially we used this one because of the small sample number, but the data is normally distributed, and so a one-way ANOVA is appropriate). From examining all the images again, we can ascertain that there is indisputably less active caspase-3 staining in *RhoGEF2-OE Ptp10D-KD* compared to *RhoGEF2-OE Dicer2*. We have selected a more suitable image that better represents this snapshot of active caspase-3 staining in *RhoGEF2-OE Ptp10D-KD*. Also, a more representative control image is now shown, where some baseline active caspase-3 staining is present.

A minor concern relates to the interpretation and consistency of the statistical analyses used. For example, in Figure 5I, both the Kruskal-Wallis test and an unpaired t-test were used, with the authors stating that the t-test was applied specifically to compare wild-type and *crb*^{-/-} clones ($p = 0.0147$). However, in the adjacent panel (Figure 5J), only a one-way ANOVA was used. This inconsistency may give the impression that the choice of statistical test in Figure 5I was influenced by a lack of significance with the Kruskal-Wallis test, rather than by experimental design. Unifying the statistical approach within related panels would improve clarity and minimize potential reader misinterpretation. Additionally, some of the statistical tests applied may not fully align with the underlying data distributions. Statistical methods used in parts of the manuscript may need to be reevaluated, and the rationale for their selection should be clarified in the text.

Response: We have checked the data carefully, plotted all the individual data sets in R, and the data is not normally distributed. Therefore, conducting a Kruskal-Wallis test is the best approach. This analysis shows that there is no significant difference between *crb*^{-/-} and *WT* in our experimental setting. However, there is a slight trend towards increased *crb*^{-/-} clone size. We have added a more detailed description of the statistical methods used in different situations in the Materials and Methods section.

In the section on how *crb*^{-/-} affects actin distribution and accumulation within the tissue (Figure 6H' and Supplementary Figure 5), it appears that F-actin may accumulate more prominently in cytoplasmic regions rather than at cell-cell junctions under *crb*^{-/-} conditions. However, due to the current level of magnification, it is difficult to determine the precise subcellular localization. Although this question is somewhat tangential to the main focus of the manuscript and not essential for publication, it could be valuable, if the authors included a few higher-magnification images showing F-actin distribution in RhoGEF2OE Dicer2, RhoGEF2OE Ptp10D KD, and RhoGEF2OE *crb*^{-/-} conditions. Including these in the supplementary figures could help clarify how actin cytoskeletal regulation is affected.

Response: We have added zoomed-in images to Figs 6G and 6H to show the effect on F-actin more clearly. It is possible that F-actin may be more prominent in the cytoplasm in *crb*^{-/-} clones, however further experiments would be needed to provide more evidence for this, which are unfortunately beyond the scope of our capabilities at this time.

In Figure 6H' the Diap1 signal in the RhoGEF2OE condition appears non-uniform, with noticeably weaker intensity on the left side of the image and stronger signal on the right. This asymmetry is not observed in the RhoGEF2OE *crb*^{-/-} condition shown in Figure 6K'. It is unclear whether this pattern reflects a biological phenomenon consistently observed in RhoGEF2OE tissues or if it might result from technical factors such as uneven mounting or imaging. To prevent potential misinterpretation, we recommend clarifying this point, providing additional representative images if available, or replacing the current image with one that more clearly reflects the typical expression pattern.

Response: We assume the reviewer means Fig 6J, and we have replaced the image with a more representative one.

In Fig. 3B', cleaved Caspase-3 appears localized to specific regions at the WT/RhoGEF2OE interface, suggesting spatial bias in Ptp10D-dependent elimination. This raises important questions about what determines regional susceptibility—are certain tissue conditions or cell states more prone to apoptosis in this context?

Figure 3 raises the question of whether RhoGEF2OE-induced, actin-deregulated clones undergo dynamic changes, such as expanding or regressing, over the course of the larval stage. Such temporal variability could influence GFP⁺ clone size and the expression of apoptotic markers like cleaved Caspase-3 and Diap1. The stated use of the L3 stage, which spans ~48 hours (Tennessee & Thummel, 2011), lacks sufficient temporal resolution. Clarifying the timing of dissection and fixation relative to clone induction would improve interpretation of clone behavior and marker dynamics.

Response: While the reviewer raises an interesting question about spatial and temporal sensitivities to apoptosis upon genetic perturbations, we have conducted all of our experiments on samples obtained from the wandering L3 stage. We have added the following text to the Materials and Methods to make it clearer: "Wandering third-instar larvae (L3) were picked for all experiments, and for each experiment all larvae were of equivalent size."

Minor comments:

GFP signal appears weaker in the wild-type group compared to experimental conditions, raising the question of whether image processing (e.g., contrast and color balance) was applied uniformly and if this difference reflects true variation in expression.

Response: Yes, images were always identically processed. We have stated in the Materials and Methods imaging section: "Laser intensity and gain was unchanged within each experimental group".

For Figures 2, 3, and 5, including representative images for each eye phenotype category would clarify the scoring criteria. In Figure 5, the use of a "2.5" category in the main figure should be explained—does it correspond to category 3 or indicate an intermediate phenotype?

Response: Apologies for this error, and thanks to the reviewer for highlighting this. The “2.5” rating was a mistake based on a previous classification scale we used, and we have changed 2.5 to 3 in the graph. We have also included a new supplementary figure explaining our rankings (Supp Fig 10).

In Figure 5I, the y-axis range (0-150%) is broader than needed; adjusting it to 0-100% would better reflect the data and improve clarity.

Response: We have edited the Fig 5I graph accordingly.

The sentence from line 343- 348 is long and challenging to follow.

Response: We have reworded the sentence.

Missing the Figure number on Line 286.

Response: We have added the Figure number.

Reviewer #2 (Significance (Required)):

Significance:

This study is well executed and rigorously addresses previously reported variations in phenotypic outcomes across laboratories. Beyond clarifying the role of Ptp10D in cell competition, the authors establish RhoGEF2 overexpression as a reliable method to induce cell competition and identify key molecular players involved in this process. This work represents a meaningful advance by introducing novel approaches and deepening understanding of known factors in clone elimination. The mosaic RhoGEF2 overexpression technique developed in this study provides a valuable tool for investigating cell-cell interactions at the tissue level, with broad applicability in basic research. This approach holds particular promise for probing.

Response: We thank the reviewer for their support of the significance and quality of our manuscript.

Original submission

First decision letter

MS ID#: jcs.264377

MS Title: Overexpression of RhoGEF2, a regulator of the actin cytoskeleton, induces cell competition dependent on Ptp10D, Crumbs, and the Hippo signaling pathway.

Authors: Natasha Fahey-Lozano; Marta Portela; John La Marca; Helena Richardson

Article Type: Review Commons Transfer

Dear Dr Richardson,

I am happy to tell you that your manuscript has been accepted for publication in Journal of Cell Science, pending standard publication integrity checks.

Thank you for sending your manuscript to Journal of Cell Science through Review Commons.